# Genomic analysis of a pre-elimination Malaysian *Plasmodium vivax* population reveals selective pressures and changing transmission dynamics

Sarah Auburn [1,2], Ernest D. Benavente [3], Olivo Miotto [2,4,5], Richard D. Pearson [2,4], Roberto Amato[2,4], Matthew J. Grigg[1,6], Bridget E. Barber [1,6], Timothy William[6,7,8], Irene Handayuni[1], Jutta Marfurt[1], Hidayat Trimarsanto[9,10], Rintis Noviyanti[9], Kanlaya Sriprawat[11], Francois Nosten [11,12], Susana Campino[3], Taane G. Clark [3,13], Nicholas M. Anstey[1], Dominic P. Kwiatkowski[2,4] & Ric N. Price[1,12]

The incidence of *Plasmodium vivax* infection has declined markedly in Malaysia over the past decade despite evidence of high-grade chloroquine resistance. Here we investigate the genetic changes in a *P. vivax* population approaching elimination in 51 isolates from Sabah, Malaysia and compare these with data from 104 isolates from Thailand and 104 isolates from Indonesia. Sabah displays extensive population structure, mirroring that previously seen with the emergence of artemisinin-resistant *P. falciparum* founder populations in Cambodia. Fifty-four percent of the Sabah isolates have identical genomes, consistent with a rapid clonal expansion. Across Sabah, there is a high prevalence of loci known to be associated with antimalarial drug resistance. Measures of differentiation between the three countries reveal several gene regions under putative selection in Sabah. Our findings highlight important factors pertinent to parasite resurgence and molecular cues that can be used to monitor low-endemic populations at the end stages of *P. vivax* elimination.

[1] Global and Tropical Health Division, Menzies School of Health Research and Charles Darwin University, Darwin, NT, 0811, Australia. [2] Big Data Institute, Li Ka Shing Centre for Health Information and Discovery, Old Road Campus, Oxford OX3 7LF, UK. [3] Faculty of Infectious and Tropical Diseases, London School of Hygiene and Tropical Medicine, Keppel Street, London WC1E 7HT, UK. [4] Wellcome Trust Sanger Institute, Hinxton, Cambridge CB10 1SA, UK. [5] Mahidol-Oxford Tropical Medicine Research Unit, Mahidol University, Bangkok 10400, Thailand. [6] Infectious Diseases Society Sabah-Menzies School of Health Research Clinical Research Unit, 88300 Kota Kinabalu, Sabah, Malaysia. [7] Clinical Research Centre, Queen Elizabeth Hospital, 88300 Kota Kinabalu, Sabah, Malaysia. [8] Jesselton Medical Centre, 88300 Kota Kinabalu, Sabah, Malaysia. [9] Eijkman Institute for Molecular Biology, Jakarta 10430, Indonesia. [10] Agency for Assessment and Application of Technology, Jl. MH Thamrin 8, Jakarta 10340, Indonesia. [11] Shoklo Malaria Research Unit, Mahidol-Oxford Tropical Medicine Research Unit, Faculty of Tropical Medicine, Mahidol University, Mae Sot, Tak 63110, Thailand. [12] Centre for Tropical Medicine and Global Health, Nuffield Department of Clinical Medicine Research Building, University of Oxford Old Road Campus, Oxford OX3 7LJ, UK. [13] Faculty of Epidemiology and Population Health, London School of Hygiene and Tropical Medicine, Keppel Street, London WC1E 7HT, UK. Correspondence and requests for materials should be addressed to S.A. (email: Sarah.Auburn@menzies.edu.au) or to R.N.P. (email: Ric.Price@menzies.edu.au)

Outside of sub-Saharan Africa, *Plasmodium vivax* has become the predominant cause of malaria[1]. Reports of antimalarial drug resistance and life-threatening complications in children and pregnant women underline the importance of containing this species[2–6]. However, whilst the incidence of *P. falciparum* has declined in most endemic countries, the proportion of *P. vivax* infections in co-endemic regions has demonstrated a consistent rise, highlighting the remarkable adaptive potential and relative resilience of this species[7]. The re-emergence of *P. vivax* in many regions following successful elimination programmes serves as an important reminder of the parasite's propensity to resurge in areas where it was once almost eliminated[8]. In order for malaria programmes to interrupt transmission successfully, a relentless focus is required on surveillance to identify foci of infection and instigate appropriate control responses to eliminate these. A better understanding of the transmission dynamics and adaptive processes of the parasite is essential. Whilst several genomic studies have explored the molecular dynamics of *P. vivax* populations with stable transmission[9–13], aside from a small selection of microsatellite-based studies[14–16], little is known about the molecular epidemiology during the critical, end stages of malaria elimination.

Malaysia has experienced a sharp decline in *P. vivax* prevalence over the past decade[17,18], providing a unique endemic setting from which to assess the changing transmission dynamics and selective pressures as the parasite population enters the vulnerable malaria pre-elimination phase. A recent clinical trial demonstrated high-grade chloroquine resistance (CQR) with parasite recurrence reported within 28 days of starting treatment in 60% of patients[19]. Despite declining treatment efficacy, the number of reported *P. vivax* cases has fallen from approximately 3000 in 2005 to under 100 in 2015[1]. Most *P. vivax* malaria is reported from the island of Borneo, where control efforts are challenged by constraints in case detection and accessibility to effective diagnosis and treatment. A recent microsatellite-based study of isolates collected in Sabah between 2010 and 2012 demonstrated local fragmentation and evidence of unstable transmission[14]. Since then, indigenous cases of *P. vivax* have continued to decline, but the threat of resurgence remains high, driven by an increase in imported cases[1], and high levels of CQR[19].

To explore the critical transition to *P. vivax* elimination, genome-wide population genetic analysis was undertaken on clinical isolates from Sabah, Malaysia and compared with isolates from western Thailand and Indonesia. In Thailand, *P. vivax* transmission is falling and 5 to 15% of patients treated with CQ have recurrent parasitaemia by day 28 (low-grade CQR)[20]. In Papua, Indonesia *P. vivax* transmission remains high and stable, with more than 50% of patients treated with CQ having recurrent parasitaemia by day 28 (high-grade CQR)[21]. The parasite population structure in Sabah was examined for evidence of unstable transmission and bottlenecking, and comparative analyses were undertaken against Thailand and Indonesia to identify genomic regions under putative selection. Further investigation of the temporal dynamics and epidemiological risk factors of specific parasite strains were investigated in a broader selection of samples using new and published microsatellite data[14] on isolates collected between 2010 and 2015.

Our results reveal that the *P. vivax* population in Sabah is highly structured with high relatedness amongst infections, declining diversity and increased inbreeding, consistent with declining but also increasingly unstable transmission. A large clonal expansion is observed, potentially reflecting the emergence of an adaptive new strain. A high prevalence of known resistance determinants is observed in Sabah, and several new signals of selection have been revealed, requiring further investigation with clinical phenotypes. These findings highlight the substantial adaptive potential in a pre-elimination *P. vivax* population, and the need for a stronger framework for molecular surveillance to track the early emergence of adaptive new strains.

## Results

**Genomic data summary.** A total of 259 independent samples had high-quality genomic data, with <5% missing genotype calls at 527,107 high-quality typeable single-nucleotide polymorphisms (SNPs). Corresponding genotype failure rates were below 5%. The high-quality samples included 51 isolates from Sabah collected between May 2011 and December 2014; 3 (5.9%) of which came from the West Coast, with the remaining 48 (94.1%) from Kudat Division. Additional high-quality samples were available from Thailand ($n = 104$) and Indonesia ($n = 104$).

**Low complexity of infection in Sabah.** Within-sample parasite diversity was characterised using $F_{WS}$ analysis, and revealed a significantly higher proportion of monoclonal infections in Sabah (84.3%, 43/51) relative to Thailand (65.4%, 68/104) and Papua Indonesia (51.9%, 54/104) (Pearson's $\chi^2$ test, $p = 0.0234$; $p = 1.8 \times 10^{-4}$), indicative of less frequent superinfection and/or co-transmission (Fig. 1). Individual $F_{WS}$ scores are presented in Supplementary Data 1.

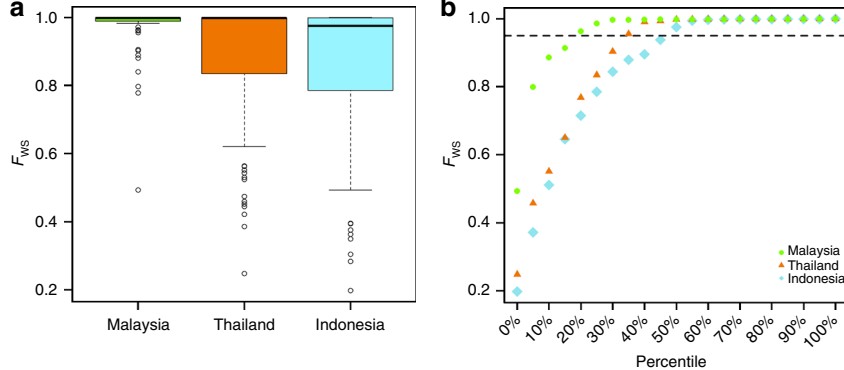

**Fig. 1** $F_{WS}$ plots illustrating trends in within-sample infection complexity in Sabah Malaysia, Thailand and Indonesia. Boxplots (**a**) and scatter plots (**b**) of the $F_{WS}$ measure, which provides a gauge of the diversity within individual infections scaled from 0 (high diversity) to 1 (no diversity). The boxplot boxes present the median and interquartile range, with the lower whiskers presenting max(min(x), quartile 1–1.5 × interquartile range). The dotted line in (**b**) illustrates the $F_{WS} = 0.95$, above which infections are essentially monoclonal. A higher proportion of monoclonal infections are observed in Sabah, Malaysia, relative to Thailand and Indonesia. The plots were generated using genomic data on 259 high-quality samples

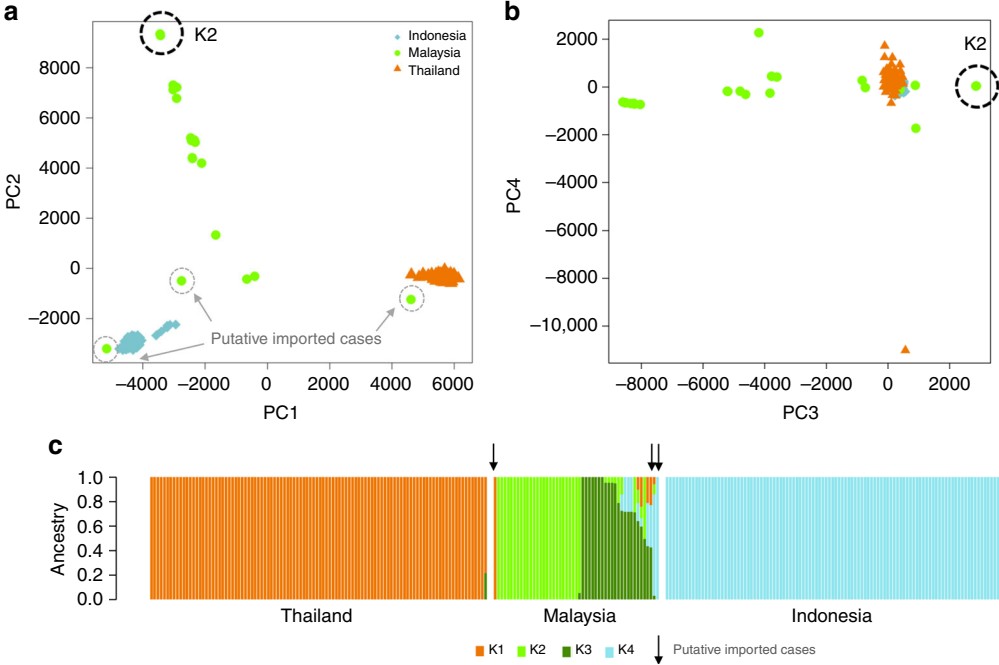

**Fig. 2** *Plasmodium vivax* population structure in Sabah relative to Thailand and Indonesia. **a, b** PCoA plots illustrating the genetic differentiation within and between populations. Principal components 1–4 reflect 17.6%, 11.7%, 3% and 1.3% of the variance, respectively. The genetically identical Malaysian K2 isolates are circled in black and the three putatively imported infections are circled in grey; the unlabelled green dots reflect the K3 and mixed ancestry infections which constitute the baseline Sabah population. Aside from the K2 strain, there is clear evidence of divergence among the baseline Sabah isolates. **c** An ADMIXTURE bar plot illustrating the population structure within and among populations at $K = 4$, highlighting the clear distinction between Thailand, Malaysia and Indonesia, and the marked sub-structure within Malaysia. All plots were generated using genomic data derived from 259 high-quality samples

**Marked population structure in Sabah.** At the population-level, principal coordinate analysis (PCoA) illustrated that, other than three putative imported cases, the isolates from Sabah were genetically distinct from Thailand and Indonesia, and exhibited striking population structure (Fig. 2a, b). Extensive population structure was observed in Sabah, in marked contrast to Thailand and Indonesia, which displayed tight clusters with no evidence of structure. Whilst the first principal component (PC1) separated Thailand and Indonesia, located over 4500 km apart, PC2 was driven by the variation within Sabah, where the maximal geographic distance between sites was <200 km. ADMIXTURE analysis confirmed the separation between countries and distinct structure within Sabah, identifying four sub-populations in the country-wide sample set (Supplementary Fig. 2). In accordance with the PCoA plots, Thailand and Indonesia formed two distinct sub-populations (K1 and K4), while Sabah was split into two major sub-populations, K2 ($n = 26$, 51%) and K3 ($n = 17$, 33%), as well as a subset of admixed samples with <70% ancestry to any given $K$ ($n = 5$, 10%), and 3 (6%) cases defined as putatively imported on the basis of ancestry ranging from 80 to 100% to K1 or K4 (Fig. 2c). Whilst the K2 sub-population was clearly diverged, there was also extensive divergence among the K3 and mixed infections. Despite the K2 and K3 isolates being derived from the same Division in Sabah (Kudat), the genetic divergence between these sub-populations ($F_{ST} = 0.479$) was greater than that observed between K1 and K4 ($F_{ST} = 0.295$).

**Genetically identical K2 infections.** Neighbour-joining analysis confirmed the extensive structure within Sabah, illustrating clusters of highly related infections, in marked contrast to more panmictic structure in Thailand and Indonesia (Fig. 3). Strikingly, all 26 K2 infections were genetically near-identical (Fig. 3). The median difference between pairs of K2 isolates across the 527,107

SNPs was five nucleotides (i.e. differing at 0.0009% SNPs on average). After excluding the imported isolates and using a single representative of the K2 strain, the median SNP-based nucleotide difference across Sabah was 11,333 (2.15% SNPs), demonstrating a moderately diverse underlying reservoir of infection despite the high prevalence of identical isolates. Higher levels of pairwise nucleotide divergence were observed in Thailand (15,433 nucleotide differences, 2.93% SNPs) and Indonesia (13,423 nucleotide differences, 2.55% SNPs).

**Large region of sequence homology in Sabah.** To explore the chromosome-level structure in the Sabah isolates, patterns of identity by descent (IBD) were explored across the genome of Sabah (both with all isolates and with a single K2 representative), Thailand and Indonesia (Supplementary Data 2). Sabah exhibited higher fractions of pairwise IBD across the genome (median = 0.446, range 0.061–0.768) than Thailand (median = 0.012, range 0.004–0.221) and Indonesia (median = 0.022, range 0.001–0.273) (Wilcoxon's rank-sum test, both $p < 2.2 \times 10^{-16}$) reflecting higher relatedness. Levels of pairwise IBD were further lower in the baseline Sabah population, where the K2 strain was represented once (median = 0.267, range 0.133–0.642). Illustrations of the IBD along each chromosome are presented in Supplementary Fig. 3. The top 5% IBD positions in Malaysia (all or baseline samples) were distributed across 10 regions on chromosomes 3, 4, 5, 8, 10 and 12. The largest stretch of IBD was observed on chromosome 12 (1942–2599 kb), with high fractions of IBD between both K2 and K3 isolates. The region encompasses 153 genes including *multidrug resistance 2* (*MDR2*, PVP01_1259100), and *gamete egress and sporozoite traversal* (*GEST*, PVP01_1258000). Notable genes in other IBD regions include *multidrug resistance 1* (*MDR1*) on chromosome 10, and *dihydrofolate synthase/folylpolyglutamase synthase* (*DHFS-FPGS,*

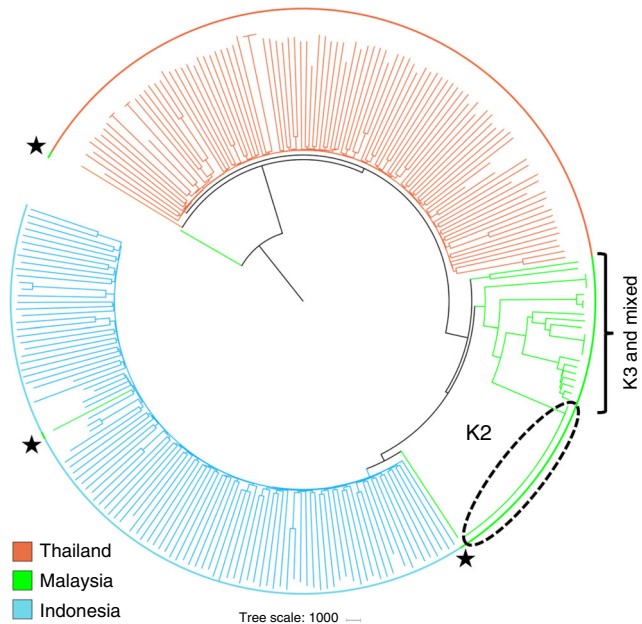

**Fig. 3** Neighbour-joining tree illustrating the relatedness between the *P. vivax* isolates from Sabah relative to Thailand and Indonesia The plot was created from genomic data on 259 high-quality infections and illustrates a rooted tree highlighting the genetically identical K2 cluster in Sabah, Malaysia. The K3 and mixed infections, which form the baseline Sabah population, are also annotated. Three putatively imported cases presenting in Sabah are annotated with black stars. One of the putatively imported infections (PY0045-C) aligned with the Papua Indonesian isolates, suggestive of importation from this region, whilst the other two cases aligned between Papua Indonesia and Sabah (PY0004-C), and close to Thailand (PY0120-C), presumably reflecting importation from regions not represented by the available sample set

PVP01_1229800) on chromosome 12. Details on the top IBD regions in Thailand and Indonesia, where a higher threshold of 1% was applied, can be found in Supplementary Fig. 3.

**High prevalence of drug resistance mutations in Sabah**. To assess whether the structure within Sabah reflected drug selection, the prevalence of mutations previously associated with clinical or ex vivo antimalarial resistance[2] was investigated (Table 1). Both the K2 and K3 sub-populations exhibited 100% prevalence of the *MDR1* F1076L and Y976F variants and the *dihydropteroate synthase* (*DHPS*) A553G and A383G variants. The prevalence of reference alleles differed between K2 and K3 at *DHPS* G626A, A619T, E618D and E132G (0% in K2 vs. 76–82% in K3, Pearson's $\chi^2$ test, $p = 4.4 \times 10^{-4} - 4.6 \times 10^{-4}$) (Supplementary Data 3); the clinical relevance of these variants is unknown. Although differences were observed in the prevalence of *dihydrofolate reductase* (*DHFR-TS*) quadruple mutants (0% in K2 vs. 67% in K3, Pearson's $\chi^2$ test, $p = 1.0 \times 10^{-5}$), overall there was a 100% prevalence of one or more mutations in both sub-populations. None of the Sabah isolates had increased copy number (CN) of *MDR1*.

The prevalence of non-synonymous variants in orthologues of genes implicated in drug resistance in *P. falciparum* was also investigated (Supplementary Data 3). *Plasmepsin IV* (PVP01_1340900) is an orthologue of *P. falciparum plasmepsin II* (PF3D7_1408000), in which CN amplification has been associated with piperaquine resistance[22,23]. The frequency of a *plasmepsin IV* I165V variant differed significantly between Sabah (90%) and Thailand (40%, Pearson's $\chi^2$ test, $p = 0.0052$), but not Indonesia (89%, Pearson's $\chi^2$ test, $p = 1.000$). There was no significant difference between the three countries in the

prevalence of variants in *pvcrt-o* (PVP01_0109300) or *kelch-13* (PVP01_1211100), whose *P. falciparum* orthologues (PF3D7_0709000 and PF3D7_1343700) are associated with CQ and artemisinin resistance respectively[24,25]. Details of the non-synonymous mutations of the *P. vivax* orthologues of *P. falciparum multidrug resistance-associated proteins 1* and *2*, and *multidrug resistance protein 2* (PVP01_020300, PVP01_1447300 and PVP01_1259100 respectively) are presented in Supplementary Data 3.

**Highly differentiated variants in drug resistance candidates**. To identify potential new modulators of drug resistance, measures of genetic differentiation ($F_{ST}$) and *Rsb*-based cross-population extended haplotype homozygosity were undertaken to detect gene regions with evidence of selection. The extensive homology among the K2 isolates obscured signal detection in Sabah; hence, analysis was undertaken with a single representative of this strain. A high-level summary of the multi-SNP $F_{ST}$ and Rsb signals is presented in Supplementary Data 4, with detailed results provided in Supplementary Data 5 and 6, respectively. Seven Sabah-specific regions exhibited multiple highly differentiated ($F_{ST} > 0.8$) SNPs (Fig. 4a, b). In Rsb-based comparisons between Sabah and each of Thailand and Indonesia, the top 0.5% of SNPs all comprised regions with extended homozygosity in Sabah (Fig. 5a, b), with 13 and 15 multi-SNP regions identified, respectively.

Evidence of differential selection was detected with the $F_{ST}$ metric at multiple SNPs in the *DHFR-TS* and *DHPS* regions. Both the $F_{ST}$ and Rsb measures found evidence of differential selection in the chromosome 12 region with high homology in Sabah. Two non-synonymous *MDR2* variants (V43L and A603T) were among the top 0.5% of SNPs identified with Rsb analysis in comparisons against Sabah. The chromosome 5 region with high homology in Sabah also demonstrated evidence of differential selection supported by the $F_{ST}$ and Rsb measures. In other genomic regions, putative drug resistance candidates included a putative *drug/metabolite transporter* (*DMT1*, PVP01_1424900) in a region of chromosome 14 supported by multiple SNPs amongst the top 0.5% of Rsb scores exhibited a V217I variant displaying high differentiation between Sabah and Thailand ($F_{ST} = 0.71$), and Thailand and Indonesia ($F_{ST} = 0.88$), but not between Sabah and Indonesia ($F_{ST} = 0.02$). Also on chromosome 14, a *CG2-related protein* (PVP01_1450700) displayed multiple SNPs amongst the top 0.5% of Rsb scores.

The putative drug resistance-associated signals in the comparisons of Thailand vs. Indonesia (Figs. 4c and 5c) have been described previously[11]. These include SNPs in *DHFR-TS* and *DHPS* demonstrating high differentiation by $F_{ST}$ analysis, and regions with extended haplotype homozygosity in Thailand encompassing *DHFR-TS* and *MRP1*. A region with extended haplotype homozygosity in Indonesia was also observed downstream of *MDR1*.

Several genes with functions unrelated to antimalarial drugs displayed highly differentiated variants, including *GEST*, which exhibited two non-synonymous mutations, E68D and K144R, displaying high differentiation between Sabah and both Thailand and Indonesia (all $F_{ST} > 0.90$), and a Q230L mutation among the top 0.5% Rsb scores in comparisons of Malaysia to Thailand. In other regions, the *merozoite surface proteins* 1 (*MSP1*) and 5 (*MSP5*) were notable, exhibiting 15 and 3 variants, respectively, among the top 0.5% Rsb scores in comparisons of Malaysia to Thailand and Indonesia.

**Signals of extended haplotype homozygosity using the iHS**. Further haplotype-based scans were undertaken using the integrated haplotype score (iHS) measure to identify regions with

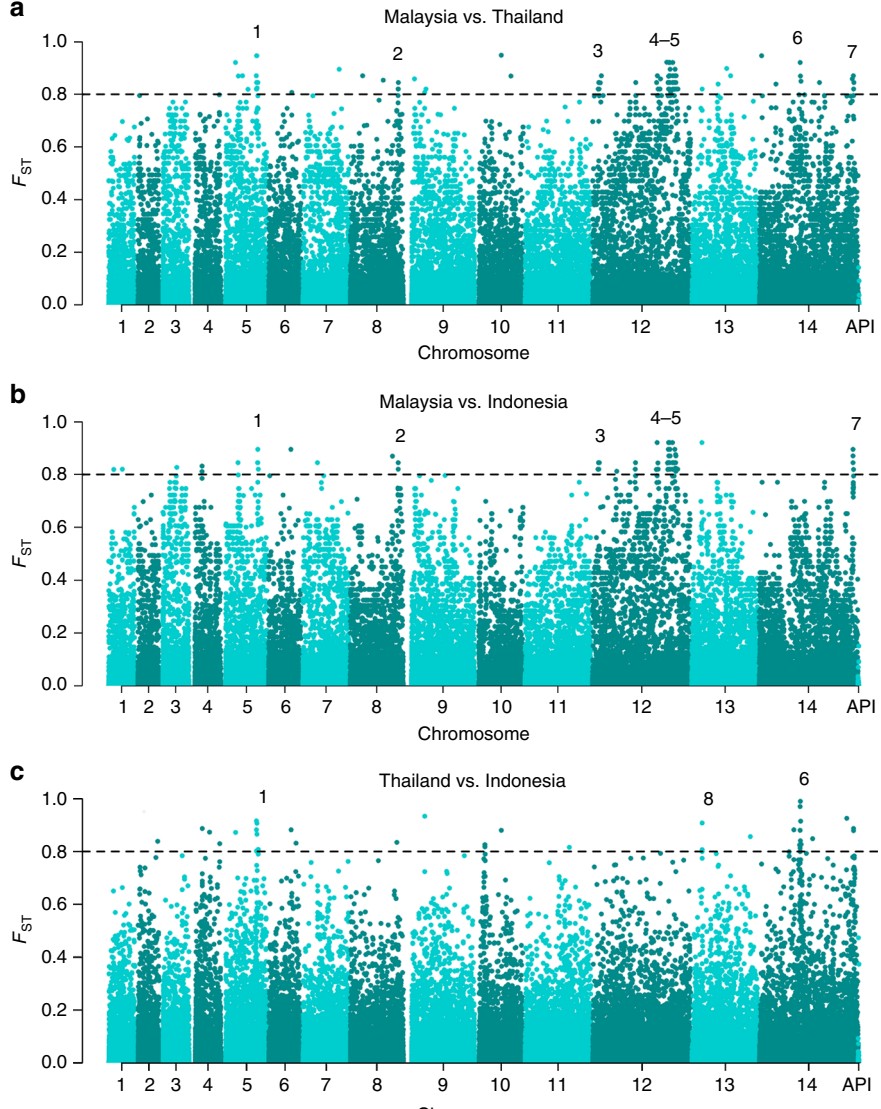

**Fig. 4** Genetic differentiation between Malaysia, Thailand and Indonesia. **a–c** Manhattan plots illustrating the $F_{ST}$ between Malaysia (Sabah), Thailand and Indonesia. The Sabah population included a single K2 representative and excluded the three imported cases. Horizontal dashed lines illustrate $F_{ST} = 0.8$. A total of 97 and 76 positions exhibited $F_{ST}$ scores ≥0.8 between Sabah and Thailand and Indonesia, respectively; 27 and 19 conferring non-synonymous changes. Seven multi-SNP regions identified in comparisons against Sabah are labelled: these regions comprise ≥3 SNPs with $F_{ST} ≥ 0.8$ within 80 kb of one another. Region 1 encompasses *DHFR-TS*. Region 2 contains two genes on chromosome 8 with unknown function (PVP01_0834900 and PVP01_0835000). Region 3 is located on chromosome 12 and includes a *nucleoside transporter* (*NT1*) with a highly differentiated *L188M* mutation between Sabah and Thailand, but not Sabah and Indonesia. Regions 4 and 5 are located adjacently on chromosome 12, corresponding with the highly homologous Sabah sequence. The regions contain biologically plausible drug resistance candidates including *MDR2* and *haeme detoxification protein* (*HDP*). Another notable gene is *GEST*, implicated in transmission between the parasite and both its *Anopheline* and human hosts[37]. Region 6 encompasses *DHPS*. Region 7 is located on chromosome 14, encompassing 7 genes but with no clear driver of selection. Region 8 contains several SNPs on chromosome 13 exhibiting differentiation between Thailand and Indonesia only. The region encompasses the *cysteine repeat modular protein 3* (*CRMP3*). In addition to *GEST* and *CRMP3*, two other genes with roles in host cell invasion or egress displayed highly differentiated variants; *CRMP1* and *glideosome-associated protein 40* (*GAP40*). Signatures of selection between southeast Asian and South American *P. vivax* populations have previously been observed in *CRMP1*[26], but this is the first report of selection in *GEST* and *GAP40*

differential levels of linkage disequilibrium (LD) surrounding the ancestral (*P. cynomolgi*) relative to the derived allele. Derived alleles could be confidently called at only 10,167 loci and, after filtering out non-informative loci, the homology in Sabah proved too extensive to permit iHS analysis in this population alone. The analysis was therefore conducted on the pooled Sabah, Thai and Indonesian samples. Four regions were supported by multiple SNPs amongst the top 0.5% of scores (Fig. 5d and Supplementary Data 4). The first region overlaps with a previously described

signal on chromosome 10, potentially reflecting selection on *MSP3.2*[26]. The second region comprised SNPs in the chromosome 12 region with high homology in Sabah. The third region presents a signal on chromosome 13, possibly driven by selection on *plasmepsin IV*. The fourth region overlaps with a previously described signal on chromosome 14 postulated to reflect selection on an *AP2 domain transcription factor* (PVP01_1418100)[13]. Further details are presented in Supplementary Data 7.

**Table 1 Prevalence of variants that have previously been associated with *P. vivax* drug resistance**

| Gene | Chr. | Positions | Mutation | Drug | Sabah frequency | Sabah K2 frequency | Sabah K3 frequency | Thailand frequency | Indonesia frequency |
|---|---|---|---|---|---|---|---|---|---|
| *MDR1* | 10 | 479,908 | F1076L | CQ | 98% (50/51) | 100% (26/26) | 100% (17/17) | 50% (54/104) | 100% (104/104) |
| (PVP01_1010900) | 10 | 480,207 | Y976F | CQ, AQ + SP | 94% (48/51) | 100% (26/26) | 100% (17/17) | 13% (14/104) | 100% (104/104) |
| | 10 | Copy number variant | 2+ copies | MQ | 0% (0/49) | 0% (0/26) | 0% (0/17) | 19% (20/104) | 0% (0/84) |
| *DHFR-TS* | 5 | 1,077,530; 1,077,532 | F57L/I | Antifolate, AQ + SP | 96% (46/48) | 100% (26/26) | 100% (15/15) | 91% (88/97) | 81% (71/88) |
| (PVP01_0526600) | 5 | 1,077,533; 1,077,534; 1,077,535 | S58R | Antifolate, AQ + SP | 34% (17/50) | 0% (0/26) | 69% (11/16) | 100% (104/104) | 99% (97/98) |
| | 5 | 1,077,543 | T61M | Antifolate, AQ + SP | 25% (13/51) | 0% (0/26) | 67% (10/15) | 91% (90/99) | 81% (71/88) |
| | 5 | 1,077,711 | S117N/T | Antifolate, AQ + SP | 25% (12/48) | 100% (26/26) | 100% (17/17) | 100% (102/102) | 99% (87/88) |
| | | | Double mutant | Antifolate, AQ + SP | 73% (35/48) | 100% (26/26) | 33% (5/15) | 1% (1/89) | 17% (15/87) |
| | | | Triple mutant | Antifolate, AQ + SP | 2% (1/48) | 0% (0/26) | 0% (0/15) | 0% (0/89) | 0% (0/87) |
| | | | Quadruple mutant | Antifolate, AQ + SP | 25% (12/48) | 0% (0/26) | 67% (10/15) | 99% (88/89) | 82% (71/87) |
| *DHPS* | 14 | 1,270,401 | A553G | Antifolate | 92% (46/50) | 100% (26/26) | 100% (17/17) | 98% (98/100) | 18% (16/91) |
| (PVP01_1429500) | 14 | 1,270,911 | A383G | Antifolate | 96% (48/50) | 100% (26/26) | 100% (17/17) | 100% (104/104) | 97% (99/102) |

Mutation prevalence was calculated with homozygous calls only. Genotype failures were <5% at all markers in all populations
*CQ* chloroquine, *AQ* amodiaquine, *SP* sulfadoxine-pyrimethamine, *MQ* mefloquine

Assessment of the iHS and of the Rsb tests using strictly monoclonal samples demonstrated no major differences in the key trends, confirming the accuracy of the results on the low-complexity samples (Supplementary Fig. 4).

**CN amplifications in Sabah**. Investigation of CN variation was undertaken on 46 Sabah, 104 Thai and 84 Indonesian isolates: excluding 22 isolates with uneven coverage. Three regions demonstrated evidence of CN amplification in Sabah; a *28S ribosomal RNA* gene on chromosome 5 (PVP01_0504500), *duffy-binding protein 1* on chromosome 6 (*DBP1*, PVP01_0623800)[11,27,28] and an *exported Plasmodium protein* with unknown function on chromosome 14 (PVP01_1470400)[11] (Table 2 and Supplementary Data 8). The PVP01_0504500 and PVP01_1470400 amplifications were highly prevalent in Sabah (82–96% in K2 and K3), and also demonstrated moderate to high frequency in Thailand (92 and 49%) and Indonesia (95 and 46%).

**Increasingly unstable transmission in Sabah over time**. To explore the epidemiology of *P. vivax* in Sabah further, genetic data were obtained on a larger collection of samples using short tandem repeat (STR) genotyping. A total of 212 independent PCR-confirmed *P. vivax* mono-species or mixed-species infections were successfully genotyped (<50% marker fails), with 187 isolates having complete data. Demographic details of the patients are summarised in Supplementary Table 1.

Among the 51 genomic samples, 47 could be genotyped successfully, with 46 having complete data. Consistent with the genomic data, all 25 K2 infections exhibited the same multi-locus genotype (MLG38). A further 44 samples carried MLG38, comprising 69 K2 isolates in total (Fig. 6a). Assessment of the temporal dynamics of the K2 strain revealed epidemic-like transmission. K2 was first observed in 2013 (29%), rose sharply in prevalence in 2014 (55%) and persisted into 2015 (83%) up to the end of the collection period (Fig. 6b).

In addition to K2, several smaller clusters of highly related infections were observed in Sabah, indicative of broadly unstable transmission in the region (Fig. 6a). Collectively, the results of several population genetic measures indicated a state of declining transmission intensity concurrent with increasing instability. First, a gradual decline was observed in the percentage of polyclonal infections, dropping from 50% in 2010 to 8% in 2014 ($p = 1.16 \times 10^{-6}$), suggestive of reduced rates of superinfection

(Supplementary Table 2), and this was associated with a strong, positive correlation between the annual number of *P. vivax* cases and the proportion of polyclonal infections (Pearson's correlation, $r = 0.98$, $p = 0.004$). Second, there was a trend of declining diversity, as measured by allelic richness ($Rs$), over the study years (Supplementary Table 2). Third, investigations of LD revealed high allelic associations ($I^S_A$ range 0.167–0.536), reflecting high rates of inbreeding owing to a composite of bottlenecking and clonal expansions (Supplementary Table 3). On restricting analysis to unique MLGs, LD remained significant in all years, but the $I^S_A$ scores dropped between 1.5-fold and 10-fold in each year, indicative of epidemic-like transmission. The trends in $Rs$ and LD remained the same when analysis was restricted to low-complexity infections (Supplementary Tables 2 and 3).

**High proportion of outbreak (K2) infections among students**. To characterise the epidemiology surrounding the expansion of the K2 strain (MLG38), we explored the demographics of individuals carrying K2 vs. low-frequency strains (Supplementary Table 4). No significant difference was observed in the proportion of males (62 vs. 73%, Pearson's chi-squared test, $p > 0.05$) or median parasite density (4180 vs. 4186 parasites μl$^{-1}$, Wilcoxon's rank-sum test, $p > 0.05$). However, a significant difference was observed in the median age of patients carrying K2 (15 years) vs. low-frequency MLGs (24 years, $p = 0.0137$). Occupational details were available for 182 (97%) of the patients with complete MLGs. Two categories were investigated; agricultural/foresty-related occupations and students. A higher proportion of students were observed among the patients carrying K2 (37%) vs. low-frequency MLGs (16%, Pearson's chi-squared test, $p = 0.012$). No significant differences were observed in the proportion of patients with agricultural/forestry-related occupations (Pearson's chi-squared test, $p > 0.05$).

**Discussion**
Multiple genomic studies have been conducted in the Asia-Pacific region[10,11,13], but none have examined the molecular epidemiology of *P. vivax* during the critical transition phase to elimination. Our investigation of the genetic and genomic epidemiology of *P. vivax* in Sabah, Malaysia, provides important insights into the molecular changes taking place in a parasite population in the late stages of malaria elimination. Our analysis reveals a highly structured population with high relatedness,

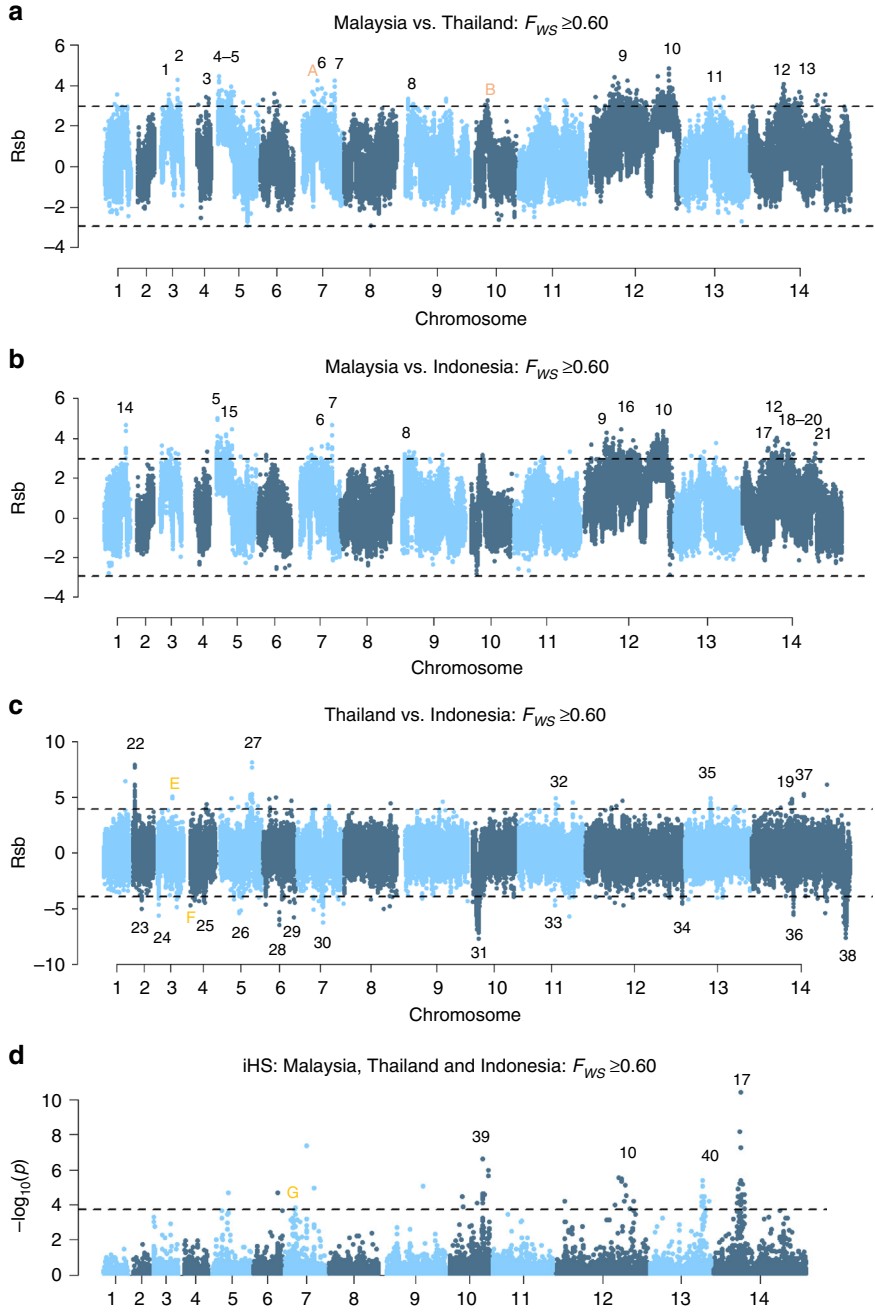

**Fig. 5** Genome-wide scans to identify regions with extended haplotype homozygosity using low-complexity samples. **a–c** Manhattan plots of the Rsb index for the given populations, and **d** the iHS *p* value for the pooled populations. Analyses were conducted on low-complexity samples ($F_{WS} \geq 0.60$). The dashed black lines demark the top 0.5% SNPs with the most significant *p* values. Regions with ≥3 SNPs within 80 kb of one another and overall SNP density <10 kb per SNP amongst the top 0.5% SNPs are numbered in black. Several previously described signals in known or putative drug resistance candidates were identified including *DHPS* (region 19), *DHFR-TS* (region 27) and *MRP1* (region 22)[10-13]. A previously described signal on chromosome 10 (region 31) is downstream of MDR1[11]. The signal in region 10 reflects the highly homologous region on chromosome 12. Putative genetic drivers in the chromosome 12 region are MDR2[13], GEST and HDP, but the latter has no directly supporting SNPs. Other drug resistance candidates include *nucleoside transporter 2* and *folate transporter 1* in regions 4 and 5, *drug/metabolite transporter 1* (*DMT1*) in region 12, *CG2-related protein* in region 21, a previously described *voltage-dependent anion-selective channel* (region 35)[11], and *plasmepsin IV* in region 40. The putative drivers in regions 3, 7, 9 and 39 include several merozoite surface proteins: *MSP5, MSP1, MSP7*-like gene cluster and *MSP3.2*, respectively[12, 26]. Another surface protein presenting a putative driver is the *duffy-binding protein* in region 29. In other gene classes, candidate drivers include *zinc finger protein* (region 6), *calcium/calmodulin-dependent protein kinase* (region 8), *filament assembling protein* (region 14), *Plasmodium exported proteins* (region 15), an *AP2 domain transcription factor* (region 17)[13], *DNA-binding protein* (region 23), *3' exonuclease* (region 25) and *JmjC domain-containing protein* (region 36). The putative genetic drivers in regions 13, 18, 24, 28, 34 and 38 have unknown function, and those in regions 1, 2, 11, 16, 20, 26, 30, 32, 33 and 37 remain unclear. Orange letters indicate regions with 1–2 SNPs amongst the top 0.5% SNPs that were represented by ≥3 SNPs in the monoclonal samples ($F_{WS} \geq 0.95$) and are detailed in Supplementary Fig. 4

**Table 2 Summary of copy number variants observed in Sabah**

| Gene | Description | Chr. | Sabah frequency | Sabah K2 frequency | Sabah K3 frequency | Thailand frequency | Indonesia frequency |
|---|---|---|---|---|---|---|---|
| PVP01_0504500 | *28S ribosomal RNA* | 5 | 89% (41/46) | 96% (25/26) | 82% (14/17) | 92% (96/104) | 95% (80/84) |
| PVP01_0623800 | *Duffy-binding protein 1* | 6 | 4% (2/46) | 0% (0/26) | 6% (1/17) | 30% (31/104) | 6% (5/84) |
| PVP01_1470400 | *Plasmodium exported protein* | 14 | 89% (41/46) | 96% (25/26) | 82% (14/17) | 49% (51/104) | 46% (39/84) |

Twenty-two samples with an excess of CNVs (18+) were excluded from the analysis

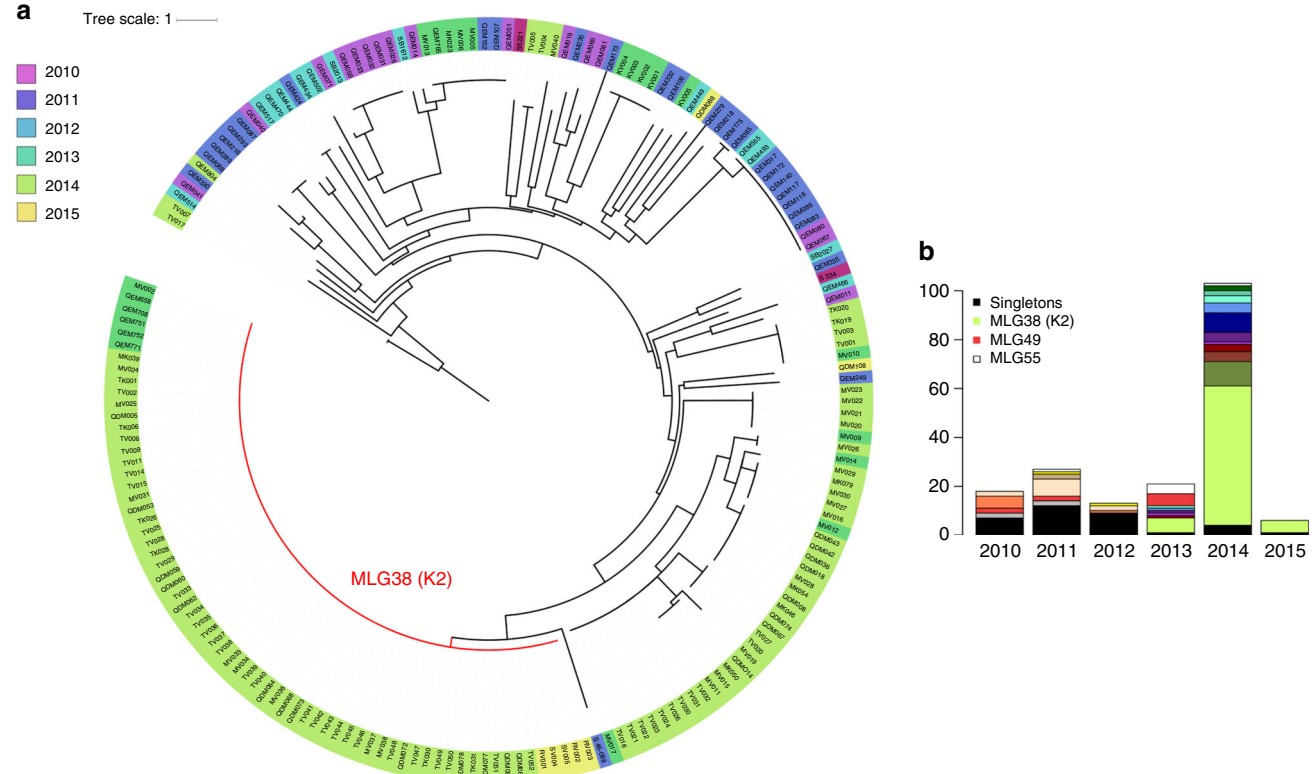

**Fig. 6** Temporal trends in the relatedness of Sabah *P. vivax* isolates using the microsatellite data. **a** A neighbour-joining tree illustrating trends in the relatedness of *P. vivax* infections between 2010 and 2015. Data are presented on 187 *P. vivax* isolates with complete MLGs across the nine microsatellite loci. **b** A bar plot illustrating temporal trends in the frequency of different MLGs between 2010 and 2015. Colour coding reflects distinct MLGs with the exception of singly observed (singleton) MLGs, which were grouped together for simple visual representation (back bars). MLG38 (genomic K2 strain) was the most frequent MLG observed in the dataset. MLGs 49 and 55 persisted for at least 4 years

declining diversity and increased inbreeding demonstrative of marked interruption to local transmission. A continual flux was observed in the genetic make-up of the parasite population, including several small and one large, rapidly emerging clonal expansion. Whilst some of these changes may have resulted from demographic or neutral processes, others may reflect more concerning adaptive changes as the parasite population is bottlenecked under heavy pressure from antimalarial drugs and other selective forces.

Antimalarial drug pressure in *P. falciparum* has been shown to have an important role in shaping the genetic make-up in Cambodian isolates[29]. We postulate that selection may have had a similar impact on *P. vivax* in Sabah. Genomic studies have shown that antimalarial drugs confer some of the strongest selective pressures on the parasite genome[10–13]. Intense drug selection is a highly likely consequence of the final stages of elimination, providing an environment in which only the most drug-resistant

infections survive[30]. Furthermore, the population structure in low transmission settings may foster the emergence of multigenic resistance phenotypes. In highly structured populations where inbreeding is common, the resistance alleles are likely to be common and unlikely to be separated during recombination. The Sabah *P. vivax* population exhibited a highly structured pattern, with divergence not just of the K2 strain, but also among the K3 and mixed infections. The extreme divergence in Sabah relative to the other Asia-Pacific populations mirrors that observed in *P. falciparum* in Cambodia as artemisinin-resistant founder populations emerged[29], suggesting that founder populations may already be emerging in response to selective pressures. However, additional phenotypic data are needed to confirm this hypothesis as the extensive divergence in Sabah may also reflect extensive bottlenecking alone.

Until recently, CQ was recommended for treating *P. vivax* blood-stage infections in Malaysia. However, the vivax population

in Sabah has also been exposed to artemisinin, mefloquine, lumefantrine and sulfadoxine plus pyrimethamine, which have been used to treat co-endemic *Plasmodium* species. In 2016, the efficacy of CQ against *P. vivax* in Sabah was shown to have fallen below 40%, and antimalarial guidelines were changed to artemether-lumefantrine for all species of malaria[19]. Prior to this date there had been no formal clinical evaluation of CQR in *P. vivax* in Malaysia, although a retrospective hospital-based study from peninsular Malaysia between 1994 and 2010 suggested low levels of CQR[31].

We were able to compare the parasite genomes collected in Sabah with those collected from western Thailand and Papua, Indonesia. Previous clinical surveys between 2007 and 2010 have shown low levels of CQR (<15% recurrence by day 28) in *P. vivax* populations on the Thai-Myanmar border[32,33]. In Papua, Indonesia high-grade CQR (>60% recurrence by day 28) was documented in 2005 following which antimalarial policy was changed to DHA-piperaquine[21]. Over the past 12 years nearly all patients with uncomplicated malaria, in both the public and private sector, are treated with DHA-piperaquine, with very little use of CQ[34]. The prevalence of CQ resistant *P. vivax* isolates may therefore have declined in recent years; however, no clinical data are available to confirm this.

We observed a high prevalence (95–100%) of polymorphisms previously associated with drug resistance in *P. vivax*, including double, triple and quadruple mutations in *DHFR-TS* and *DHPS* in Sabah. Signals of differential selection reflecting different haplotypic backgrounds were also observed in the *DHFR-TS* and *DHPS* regions in comparisons between Sabah and Thailand and Indonesia. A high prevalence of *MDR1 Y976F* and *F1076L* variants was also observed in Sabah. However, *MDR1* appears to be, at best, a modulator of CQR, with definitive markers of resistance to this drug remaining elusive[2].

On investigation of polymorphisms previously associated with drug resistance in *P. vivax* in K2 vs. K3, we observed differences in the prevalence of several *DHPS* and *DHFR-TS* alleles, potentially reflecting sulfadoxine-pyrimethamine pressure. The extensive homology in the K2 sub-population did not permit genome-wide investigations of differential selection and haplotype homozygosity to identify any novel selective determinants. Nonetheless, although the K2 sub-population was notable for its extreme genetic identity, the extensive divergence across Sabah was not restricted to this strain. Therefore, rather than limiting our investigation of drug-related selective pressure to the K2 strain, we undertook genome-wide tests of selection in the baseline Sabah population using a single K2 representative.

Several new genes identified using genome-wide tests of selection in the current study present plausible molecular determinants of CQR. The *MDR2* gene is a biologically plausible resistance determinant owing to its function as a cellular transporter, and was located on a specific haplotype background in the majority of Sabah isolates in a region with evidence of extended haplotype homozygosity based on iHS analysis, and differential selection based on $F_{ST}$ and Rsb measures. The *MDR2* gene also demonstrated differential selection in Sabah relative to Thailand and Indonesia at non-synonymous variants supported by both the $F_{ST}$ and Rsb metrics.

The *P. vivax CG2-related protein*, on chromosome 14, presents another putative determinant of CQ efficacy. In *P. falciparum*, *CG2* protein is involved in the uptake and digestion of haemoglobin in erythrocytic stage parasites[35]. Other biologically plausible genes identifed as potential modulators of resistance to CQ or other antimalarials on the basis of intragenic supporting SNPs with evidence of genetic differentiation or extended haplotype homozygosity included *plasmepsin IV* and *drug/metabolite transporter 1*. The *plasmepsin IV* gene on chromosome 13, which

encodes a hemoglobinase, and exhibited an *I165V* variant amongst the top 1% of iHS scores, is intriguing owing to the role of CN variation in *P. falciparum plasmepsin II* in piperaquine resistance[22,23].

The candidates identified add several new genes for consideration in future molecular studies of *P. vivax* drug resistance, although these require further investigation and validation with clinical or ex vivo phenotypes. Whilst we were conservative in our selection of regions under positive selection, and identified signatures in previously reported regions, the population structure in Sabah presents a challenge in this low-endemic setting. Furthermore, comparison with the Indonesian isolates, where high-grade CQR has been reported previously, may have been confounded by the high rate of local transmission[36], which may have rapidly reduced haplotype homozygosity surrounding any drug resistance alleles in this population.

Selective forces other than antimalarial drug pressure may have also shaped the genetic structure in Sabah. There was evidence of differential selection in multiple genes involved in gamete egress and sporozoite traversal, including *GEST*. The *P. berghei* orthologue of *GEST* has important roles in mediating transmission between the parasite and both its *Anopheline* and human hosts[37], and hence certain variants of this gene may influence transmission. In addition, there was evidence of extended haplotypes in Sabah in regions encompassing *MSP1* and *MSP5*, potentially reflecting immune-related pressure.

In addition to selective forces, the genetic dynamics in Sabah are likely to have been influenced by demographic factors. Using STR-based genotyping data, we were able to explore the contribution of several host demographic factors to the expansion of the K2 strain. The data revealed that students relative to other occupational groups assessed were more likely to carry K2 alleles than low-frequency strains. This finding might reflect factors related to the immune status of this particular age group, who had a median age of 15 years. Alternatively, it might reflect a high-frequency of K2 transmission events taking place in or around school, college or home, highlighting occupation as a potentially important epidemiological link in reactive case detection strategies. However, further investigation of transmission networks is required with more detailed epidemiological data.

Based on clustering patterns in the genomic data, we were able to determine putative imported infections, identifying three cases in Sabah. The K2 strain did not display any evidence of importation. Rather, the large stretches of homologous sequence between K2 and other Sabah infections in regions such as chromosome 12 suggests recent common ancestry. However, we cannot discard the possibility that the K2 lineage derives from a local recombination event between an isolate from Sabah and an imported infection, bringing together genetic features that were highly adaptive to the local environment.

The moderate diversity in the baseline Sabah population supports the local adaptive potential. Aside from imported infections, it will be important to determine the contribution of a hidden reservoir of asymptomatic or subpatent infections in maintaining local diversity, and whether the genetic structure of these infections is comparable to the clinical cases. A genetic study of actively and passively detected *P. vivax* cases from Timika, Papua Indonesia, revealed similar genetic diversity between the two reservoirs, although a higher frequency of polyclonal infections in the actively detected cases[38]. However, there are no data available on the genetic make-up of passively vs. actively detected *P. vivax* in low-endemic settings.

Rebound or resurgence of infection remains a great concern for malaria elimination programs. The continual flux in the genetic make-up of the Sabah vivax population, rapid expansion of new strains and high prevalence of known resistance-associated

determinants highlight the strong adaptive potential in a population on the verge of elimination. These findings have important implications for other vivax endemic countries striving to achieve malaria elimination, underlining critical areas for further research. A stronger foundation for molecular surveillance is needed, to track the early emergence of adaptive parasites that will guide appropriate and timely public health interventions.

## Methods

**Ethics**. All samples were collected with written informed consent from patients or their legal guardians. Ethical approval was provided by the Human Research Ethics Committee of Northern Territory Department of Health and Families (HREC-2010-1431, HREC-2012-1815 and HREC-2010-1396), the Medical Research Ethics Committee, Ministry of Health, Malaysia (NMMR-10-754-6684, NMRR-12-511-12579), the Mahidol University Faculty of Medical Technology (MUTM 2011-043-03), the Eijkman Institute Research Ethics Committee (EIREC-47) and the Oxford Tropical Research Ethics Committee (OXTREC-45-10).

**Study site**. The focal study site was in Sabah, Malaysia (Supplementary Fig. 1a). Details on the epidemiology of malaria in Sabah have been published previously[17]. Four species of malaria are endemic in the region: *P. falciparum*, *P. vivax*, *P. malariae* and *P. knowlesi*. Whilst the incidence of *P. vivax* and *P. falciparum* has declined substantially over the past decade, *P. knowlesi* cases have risen markedly, to become the most common cause of malaria (Supplementary Fig. 1b). There is no significant seasonal variation in malaria transmission, with the region experiencing moderately constant temperature and high humidity throughout the year. Until 2016, the national policy for treating uncomplicated *P. vivax* infection was CQ plus primaquine (14-day regimen), but following documentation of high-grade CQR, first-line treatment was changed to artemether-lumefantrine plus primaquine.

**Sample details**. Details of the genomic samples are presented in Supplementary Data 1. The Sabah samples were sourced from clinical and epidemiological studies conducted at the Queen Elizabeth Hospital in Kota Kinabalu, Kudat District Hospital or Kota Marudu District Hospital between September 2010 and June 2015[19,39,40]. Two to five millilitres of venous blood was collected from symptomatic patients with *Plasmodium* spp.-positive thick blood smears. A 200 μl aliquot of blood was set aside for PCR-based assays. The remaining aliquot was subject to leukocyte depletion by cellulose filtration in preparation for whole-genome sequencing[41]. DNA extraction was performed using commercial kits (Qiagen). *Plasmodium* species was confirmed by PCR, using previously described assays[42,43]. For comparative analysis, previously published genomic data were derived from *P. vivax* isolates from Thailand and Indonesia[11].

**Whole-genome sequencing and data analysis**. Leukocyte-depleted samples with ≥50 ng DNA comprising <90% human DNA were subject to whole-genome sequencing, read alignment and variant calling within the framework of a community study in the Malaria Genomic Epidemiology Network (MalariaGEN)[44]. Briefly, sequencing was undertaken on the Illumina GAII or Hi-Seq 2000 platform at the Wellcome Trust Sanger Institute. Paired-end libraries were prepared according to the manufacturer's protocol. Cluster generation and sequencing were undertaken following the manufacturer's protocols for generating standard paired-end 75–150 bp reads. Reads aligning to the human reference genome were removed before any analyses were undertaken. The remaining reads were aligned against the *P. vivax* P01 reference[45] using bwa-mem version 0.7.15 with -M parameter[46]. Improvements in the bam files were undertaken using Picard version 2.6.0 tools CleanSam, FixMateInformation and MarkDuplicates, and GATK version 3.6 base quality score recalibration. Standard alignment metrics were generated for each sample using the flagstat utility from samtools version 1.2 and GATK's CallableLoci version 3.5[47,48]. SNP discovery and genotype calling were undertaken using methods defined previously for *P. vivax* field isolates[11], with data derived from the MalariaGEN *P. vivax* Genome Variation Project release 3.0. A set of 532,751 high-quality bi-allelic SNPs with VQSLOD score >3 were derived from an initial set of 4,084,419 discovered variants. High-quality samples were defined as those with ≥95% calls at the high-quality SNPs. A set of 527,107 high-quality typeable SNPs was then derived by excluding SNPs with >5% missing calls in the high-quality samples. Missing calls were defined as positions with <5 reads. For haplotype-based analyses, either the major allele (highest read depth) or reference allele (at positions with equal allele depths) were used to reconstruct haplotypes at heterozygote positions. Where missing data were not permitted, haplotypes were reconstructed using information from calls with <5 reads.

For assessment of parasite relatedness, neighbour-joining (NJ) and PCoA were conducted using a pairwise distance matrix calculated using the R statistical package. The NJ plot was created using the iTOL software[49]. Assessment of population structure was performed using ADMIXTURE version 1.3.0, with the most likely number of sub-populations ($K$) determined by the cross-validation error[50]. ADMIXTURE was also used to determine the genetic differentiation ($F_{ST}$) (combined across all SNPs) between the derived sub-populations.

Within-host infection complexity was assessed using the within-sample $F$ statistic ($F_{WS}$)[51,52], which estimates the fixation of alleles within each infection relative to the diversity observed in the total population on a scale from 0 to 1. Previous studies have demonstrated that an $F_{WS} ≥0.95$ is highly indicative of clonal infection[51,52]. The significance of differences in the proportion of infections with $F_{WS} ≥0.95$ between populations was assessed using Pearson's $\chi^2$ test.

Measures of pairwise IBD were undertaken using hmmIBD with the default parameters for recombination rate (based on *P. falciparum* estimates), and genotyping error rate, and using allele frequencies estimated by the program[53]. The pairwise IBD outputs were then determined at 1 kb intervals across the accessible regions of each chromosome and average fractions of IBD derived for each position in each population. Differences in the median IBD between populations were determined using the Wilcoxon's rank-sum test.

Pairwise measures of the genetic differentiation ($F_{ST}$) at individual SNPs were calculated using Weir and Cockerham's formula using the R software[54], and results illustrated as Manhattan plots implemented with the qqman package. Analyses of cross-population haplotype diversity and genetic differentiation were restricted to high-quality typable SNPs with minor allele frequency >1% across Sabah, Thailand and Indonesia.

Population-based measures of haplotype diversity were used to scan the genome for regions under putative positive directional selection[55–57]. The Rsb measure of cross-population extended haplotype homozygosity and the iHS were measured using the R-based rehh package[58]. For iHS analysis, ancestral alleles were derived by mapping *P. cynomolgi* reads[59] against the *P. vivax* P01 reference. Only positions where *P. cynomolgi* calls were homozygote and matched either the reference or alternative *P. vivax* allele were included in analysis.

Large CN variations (CNVs) were detected using pysamstats (http://github.com/alimanfoo/pysamstats), which we have previously used and validated in a study of *P. vivax* MDR1 CNV[60]. For each sample, coverage in non-overlapping 300 bp bins was calculated and normalised by dividing by the median coverage across all bins with the same integer percentage GC content. A hidden Markov model implemented with the python package sklearn.hmm.GaussianHMM was used to call individual CNVs, and all variants >3 kb were recorded. Further validation of CN amplifications was undertaken using face-away mapping reads[27,28].

Aside from the $F_{WS}$ and $F_{ST}$ tests, all genomic analyses were undertaken using the major allele call at heterozygous positions. IBD analysis was undertaken on monoclonal samples ($F_{WS} ≥0.95$). The Rsb and iHS tests were undertaken on low-complexity samples (defined as $F_{WS} ≥0.6$), and additionally assessed with monoclonal samples ($F_{WS} ≥0.95$). All other genomic analyses were conducted without $F_{WS}$ filters.

**STR genotyping and data analysis**. Genotyping was undertaken at nine STR markers (Pv3.27, msp1F3, MS1, MS5, MS8, MS10, MS12, MS16 and MS20) using methods described previously[61]. The labelled PCR products were sized by denaturing capillary electrophoresis on an ABI-3100 Genetic Analyser with GeneScan LIZ-600 (Applied Biosystems) internal size standards. Genotype calling was undertaken using VivaxGEN version 1.0[62]. A threshold of 100 relative fluorescence units was applied for peak detection, and minor alleles were called if they had ≥33% height of the predominant allele.

The multiplicity of infection (MOI) for a given sample was defined as the maximum number of alleles observed at any of the nine markers. Population-level diversity was assessed using allelic richness ($Rs$)[63]. In the isolates with complete genotyping data, MLGs were reconstructed from the predominant alleles for assessment of the genetic relatedness between sample pairs using (1-$ps$) as a measure of genetic distance[64]. An unrooted neighbour-joining tree was generated with the R-based ape package[65]. The MLGs were also used to assess multi-locus LD. LD was measured using the standardised index of association ($I_A^S$), with significance estimates determined using 10,000 random permutations[66]. Apart from measures of polyclonality and MOI, only the predominant allele at each locus in each isolate was included in th analysis[67]. $Rs$ and LD analyses were conducted on all infections and additionally in low-complexity samples (maximum 1 multi-allelic locus) to assess the potential impact of MLG reconstruction errors.

Differences in proportions were examined using Pearson's $\chi^2$ test. Pearson's product–moment correlation and Spearman's rank correlation was used to test correlations in parametric and non-parametric data, respectively. All tests were performed using R, and assuming a significance threshold of 0.05.

**Data availability**. The Illumina sequence reads generated on the new *P. vivax* samples from Sabah, Malaysia has been deposited in the European Nucleotide Archive (ENA) under the accession codes listed in Supplementary Table 5A vcf file containing the genotyping data used in the study is available on the MalariaGEN website (https://www.malariagen.net/resource/24). The authors declare that all other data supporting the findings of this study are available within the article and its Supplementary Information files, or are available from the authors upon request.

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

## Acknowledgements

We thank the patients who contributed their samples to the study, and the health workers and field teams who assisted with the sample collections. We also thank the staff of the Wellcome Sanger Institute Sample Logistics, Sequencing, and Informatics facilities for their contribution. We thank the Director General of Health, Malaysia, for permission to publish this study. Financial support for the study was provided by the Wellcome Trust (Senior Fellowship in Clinical Science awarded to R.N.P., 200909), the National Health and Medical Research Council, Australia ('Improving Health Outcomes in the Tropical North: A multidisciplinary collaboration 'HOT North' Career Development Fellowship awarded to S.A. (1131932)) and the Bill and Melinda Gates Foundation (OPP1164105). The patient sampling and metadata collection was funded by the Asia-Pacific Malaria Elimination Network (108-07), the Malaysian Ministry of Health (BP00500420) and the Australian National Health and Medical Research Council (1037304 and 1045156; Fellowships to N.M.A. (1042072 and 1135820), B.E.B. (1088738) and M.J.G. (1074795)). M.J.G. was also supported by a 'Hot North' Early Career Fellowship (1131932). The whole-genome sequencing component of the study was supported by grants from the Medical Research Council and UK Department for International Development (M006212) and the Wellcome Trust (206194, 204911) awarded to D.P.K. T.G.C. is supported by the Medical Research Council UK (MR/K000551/1, MR/M01360X/1, MR/N010469/1, MC_PC_15103). S.C. is funded by the Medical Research Council UK (MR/M01360X/1, MC_PC_15103).

## Author contributions

S.A., E.D.B., O.M., R.D.P., R.A., F.N., S.C., T.G.C., N.M.A., D.P.K. and R.N.P. contributed to study design and management. M.J.G., B.E.B., T.W., I.H., J.M., R.N. and K.S. carried out field and laboratory work to obtain *P. vivax* samples, clinical metadata and microsatellite genotyping data. O.M., R.D.P., R.A. and D.P.K. managed the genomic data production. S.A., E.D.B., R.D.P., R.A. and H.T. performed data analyses. S.A. and R.N.P. wrote the first draft of the manuscript, and all authors contributed to and approved the final manuscript.

## Additional information

**Competing interests:** The authors declare no competing interests.

