## [Peer Review File · Nature Communications]

Reviewers' comments:

Reviewer #1 (Remarks to the Author):

This manuscript describes a population genomic analysis of *Plasmodium vivax* in three Asian populations. In one of these populations (Sabah, Indonesia), the contracting parasite population becomes dominated by a single clone (K2) and exhibits reduced haplotypic diversity in regions of chromosome 5 and 12, leading the authors to speculate that drug selection is operating in those regions. The data appear to be high quality, and the analyses have generally been expertly conducted. The findings are interesting because they enable comparison to genomic profiles that have been previously published for *Plasmodium falciparum* populations approaching elimination.

While the general observations of the manuscript are sound, several points could be more thoroughly explored or benefit from clearer interpretation. Some patterns may have alternate interpretations, or could be more accurately described using alternative terminology. Most importantly, it is difficult to assess the relative importance of neutral factors (bottlenecks) vs drug selection for the observed patterns in Sabah; a parasite population rapidly shrinking in size might very well display the same patterns of diminishing clonal and haplotypic diversity. In the absence of more convincing targets of selection in the homozygous regions on chromosomes 5 and 12, it could be more appropriate to refrain from terminology suggesting positive selection (eg 'clonal expansion').

Specific comments:

1) Abstract: "Sabah displayed extensive population fragmentation". This sentence is misleading, and could be better phrased as 'displayed decreasing haplotypic/clonal diversity' or similar. The authors do not specify the locality of samples below the level of country/island (Sabah), and 'fragmentation' to my ear connotes spatial vicariance, which is difficult to assess here. Figure 7b suggests that the K2 lineage came to predominate in Sabah between 2012 and 2015. Similarly, the phrase 'clonal expansion' in the abstract and throughout the manuscript has questionable applicability here, if the predominance of K2 came about as much through bottlenecks and population reduction as positive selection.

2) Line 163: Has pysamstats been previously used to detect CNVs in *P. vivax*? Have the authors used qPCR or an alternate approach to validate any of the CNV predictions made here?

3) Line 205: And/or less frequent co-transmission? How does Fws relate to MOI as separately determined?

4) Line 230: What does 1.81% SNPs difference mean here? Is that the median SNP difference % for comparisons that do not involve K2 isolates? Or including them?

5) Line 238/Fig 4: Would the haplotype signatures on chromosomes 5 and 12 be clearer using an identity-by-descent-based metric (eg IsoRelate/hmmIBD) rather than pi? Pi is a

continuous statistic, with its mean value here determined by the proportion of comparisons between identical vs different haplotypes, but really what the authors are differentiating are genomic regions that show an enhancement of virtually perfect identity among samples due to recent common ancestry, from those that do not. π is a less elegant way of getting at this, which could miss IBD blocks that are smaller or less frequent.

6) Line 284: How are the estimates of F_{st} and R_{sb} impacted by the high prevalence of the K2 clone in Sabah/Malaysia? Is a Weir and Cockerham estimator of F_{st} appropriate here given unequal sample size (Malaysia \ll Thailand/Indonesia)? Are the MDR2 variants still significant in a comparison of Thailand to Indonesia, leaving out Malaysia?

7) Assuming that *cynomolgi* here is assumed to be the ancestral rather than derived allele.

Figure 3: The rooted tree more clearly shows the K2 clone. Necessary to include the unrooted tree? That one appears redundant given that the other Malaysian clones besides K2 are visible in panel a.

Reviewer #2 (Remarks to the Author):

Thank you for the opportunity to review the manuscript entitled "Genomic analysis of *Plasmodium vivax* in Malaysia reveals selective pressure in a parasite population approaching elimination" by Auburn et al. This manuscript describes the population genetic analysis of 51 *P. vivax* parasite genomes from Sabah, Malaysia in comparison with genomic data from 104 *P. vivax* samples from each of western Thailand and Papua, Indonesia. The data reveal that just over half (54%) of the *P. vivax* samples from Malaysia are essentially genetically identical, perhaps representing a recent clonal expansion; and that genomic diversity reveals patterns of selection, possibly as a consequence of antimalarial drug use. Overall, this is a nice descriptive study of the *P. vivax* parasite population structure in a declining transmission context where chloroquine resistance (CQR) is relatively high. The major concern with the manuscript is the potential over-interpretation that signatures of selection are the consequence of drug pressure, given a lack of phenotypic or validation data. Nevertheless, the work represents a good description of *P. vivax* parasite population structure in a setting where transmission is declining that can inform potential challenges for elimination of *P. vivax*.

Major Comments:

I would encourage the authors to avoid potential conflation of "selection" and "clonal expansion" to infer that clonal expansion is a consequence of drug pressure.

- The large proportion of "clonal" (K2) parasites is consistent with a recent clonal expansion (based upon ISA data).
- There is evidence of genetic loci (e.g., genomic regions on chromosomes 5 and 12), but these are found in both the K2 and K3 Malaysian populations. Thus, these regions are not likely to harbour variants that selectively contribute to the clonal expansion of the K2 lineage.
- Only dhps and the quadruple dhfr-ts haplotype differs between the K2 and K3 populations

among known drug resistant loci tested; and, other regions under selection do not explain a possible "recent clonal expansion" of K2, especially since these mutations are enriched in the K3 parasite population.

- If drug resistance were the selective force driving expansion of the K2 parasite lineage, you would expect to see increases in the allele frequency of those markers over time, or association of specific variants with a drug resistant phenotype. You might also expect this lineage to be further increased in 2014, but peak seems to be 2013 by STR analysis.
- There is evidence of selection; but, I am not convinced that there is evidence that drug resistance is driving this lineage. For example, perhaps the K2 lineage is more easily able to be transmitted.

I would encourage the authors to provide more comprehensive information on polyclonal and monoclonal samples; how they were handled in the analysis; and to clarify the potential sources of bias in how polyclonal infections were handled.

- FWS is used to identify monoclonal infections. The authors should clarify the comparison used to derive the p value (0.0234, line 204) provided to support that monoclonal infections were enriched among Malaysian samples
- The authors argue that more monoclonal infections equate to "less frequent superinfection", but polyclonal infections can come from superinfection, co-transmission, relapse, or a combination thereof.
- Please specify how polyclonal infections were addressed in the analysis and how any potential bias toward relatedness was considered in the analysis.
- For example, in the STR analysis, only the dominant allele was used to create the MLG.

Analysis to quantify parasite genetic relatedness (done by neighbour-joining and PCoA) could be enhanced by testing for identity by descent (IBD) analysis that can quantify both the proportion of the genome that is related as well as the length of the chromosomal segments that are identical by descent to infer relatedness.

Interpretation that K2 infections were increased among student population, but perhaps this is an age-related effect (i.e., immunity) since students are likely enriched among younger aged individuals.

Minor Comments:

1. Indicate that 3 parasites are "imported": (a) state how imported was determined/defined (i.e., travel history); (b) identify these imported infections on analysis (i.e., are they distinct in the PCoA and ADMIXTURE analysis); and, (c) do they align with other *P. vivax* populations consistent with importation? For example, there seem to be at least two Malaysian samples that cluster with the Thailand or Indonesian parasites (Figure 2a), are these the imported parasites or parasites from the "west coast" rather than "Kudat"?
2. For PCoA, please specify how much of the variance of the data is explained by the components.
3. Define what is meant by "high grade" and "low grade" CQR.
4. You indicate the divergence between K2 and K3 is greater than the divergence between K1 and K4 (by FST), is that what you would expect given population fragmentation within Malaysia?

5. Infections from PCD, comment on how these may (or may not) represent the parasite population given so few cases. Could reservoir infections be different and clonal parasites be found only among those who are symptomatic?
6. Include reference for Rsb measure of cross-population extended haplotype homozygosity (Tang, PMID: 17579516)
7. The descriptions regarding analysis for both known drug resistance loci and those identified through selection analysis are quite wordy and difficult to follow. Perhaps including a summary table with the high level findings would help the reader.

Figure Comments:

1. Supplemental Figure 1a: there are no regions on the map that are light grey (representing "P. vivax free"); and, no regions indicated by striped grey (representing "Unstable transmission and high Duffy negativity").
2. Supplemental Figure 1b: state that "incidence of P. vivax has declined substantially over the past decade", but would be better to qualify that based upon Supplemental Figure 1b showing for cases reported that there was not much difference between 2003 and 2015 endpoints, and that there was evidence of an increase in cases between 2005 and 2010, followed by a decrease in cases from 2010 to 2015. It would be valuable to include prevalence or incidence data for these periods, if available, for context. Do these trends correspond to the decline in polyclonal infections over time?
3. More information about the details of the figures should be provided in the legends. For example, Figure 1 does not describe the information in the figure adequately and only provides a high level summary.

Discussion:

1. Clarify comment: "...continual flux was observed in the genetic make-up of the parasite population, including several rapidly emerging clonal expansions", given there was one clonal population that was consistent with rapid emergence.
2. The authors speculate that heavy drug pressure (and other selective forces) have "bottlenecked" the population, leaving the fittest isolates remaining. There is no evidence that the "fittest" have survived...and if they are indeed due to drug resistance than they may be survivors because of that phenotype. Many drug resistant parasites are actually "less fit" than their wild-type counterpart.
3. The authors postulate that drug pressure has shaped the population...would have been nice to see that drug resistance or other loci are enriched in the K2 population that has expanded. It is fine to speculate, but just be sure to qualify statements appropriately.
4. "Intense drug selection is an inevitable consequence of the final states of elimination, providing an environment in which only the most drug-resistant infections survive", while this may be the case in some geographic settings (depending upon drug pressure etc), it is not necessarily "inevitable".
5. MDR2 is focused on as a possible driver...in Ch12 region, but how does this relate to population structure of K2? I would be careful not to over interpret the identification of new candidates without any phenotypic or validation information. Fine to put them out there, and discuss how they might play a role, but not sure there is strong evidence to

suggest they are playing a role in the population structure observed.

6. Caution around the interpretation of students, as age may be the important factor and possibly related to changes in immunity. Arguments about risk populations might be addressed by spatial or temporal data, should this be available.

Reviewer #1

General comments

This manuscript describes a population genomic analysis of Plasmodium vivax in three Asian populations. In one of these populations (Sabah, Indonesia), the contracting parasite population becomes dominated by a single clone (K2) and exhibits reduced haplotypic diversity in regions of chromosome 5 and 12, leading the authors to speculate that drug selection is operating in those regions. The data appear to be high quality, and the analyses have generally been expertly conducted. The findings are interesting because they enable comparison to genomic profiles that have been previously published for Plasmodium falciparum populations approaching elimination.

While the general observations of the manuscript are sound, several points could be more thoroughly explored or benefit from clearer interpretation. Some patterns may have alternate interpretations, or could be more accurately described using alternative terminology. Most importantly, it is difficult to assess the relative importance of neutral factors (bottlenecks) vs drug selection for the observed patterns in Sabah; a parasite population rapidly shrinking in size might very well display the same patterns of diminishing clonal and haplotypic diversity. In the absence of more convincing targets of selection in the homozygous regions on chromosomes 5 and 12, it could be more appropriate to refrain from terminology suggesting positive selection (eg 'clonal expansion').

We thank the reviewer for their positive feedback and constructive suggestions. We acknowledge that some of the patterns of diversity and structure observed in Sabah may reflect processes other than selection from drugs or other forces. We have stated this in the opening paragraph of the Discussion as well as in the paragraph discussing our assessment of demographic risk factors of K2 carriage. As described in response to the specific comments below, we have now added further clarification in various parts of the manuscript.

Specific Comments

1. Abstract: "Sabah displayed extensive population fragmentation". This sentence is misleading, and could be better phrased as 'displayed decreasing haplotypic/clonal diversity' or similar. The authors do not specify the locality of samples below the level of country/island (Sabah), and 'fragmentation' to my ear connotes spatial vicariance, which is difficult to assess here. Figure 7b suggests that the K2 lineage came to predominate in Sabah between 2012 and 2015. Similarly, the phrase 'clonal expansion' in the abstract and throughout the manuscript has questionable applicability here, if the predominance of K2 came about as much through bottlenecks and population reduction as positive selection.

We agree that the terms used may be ambiguous and thus have replaced the term "fragmentation" with "structure" throughout the manuscript. We have omitted the text "indicative of a clonal expansion" on lines 127-128 because at this point in the manuscript we have not yet provided information on the changing prevalence of K2. However we would like to retain the subsequent reference to "clonal expansion" in other parts of the text where the data provided strongly supports this dynamic. In the STR genotyping results we state a 1.5 to 10-fold drop in the ISA in each year after adjusting for replicated MLGs is indicative of epidemic-like transmission. Furthermore, in the specific case of the K2 strain, the changing frequency as determined by STR genotyping from 0% in 2010, 2011,

and 2012 to 29% in 2013, 55% in 2014 and 83% in 2015 is highly consistent with a clonal expansion. To clarify this further we have added further details on the percentage of K2 strains in each year:

Lines 267-269 “K2 was first observed in 2013 (29%), rose sharply in prevalence in 2014 (55%) and persisted into 2015 (83%) up to the end of the collection period (Figure 6b).”

2. Line 163: Has pysamstats been previously used to detect CNVs in *P. vivax*? Have the authors used qPCR or an alternate approach to validate any of the CNV predictions made here?

We did not conduct PCR-based validation of all of the CNVs identified in the study. However, we have previously used pysamstats in our investigation of *P. vivax* MDR1 CNVs (Auburn *et al.*, JID, 2016 [27456706]), where we confirmed the variants using break-point PCR methods. A comment has been added to the text to clarify this point:

Lines 509-511 “Large copy number variations (CNVs) were detected using pysamstats (<http://github.com/alimanfoo/pysamstats>), which we have previously used and validated in a study of *P. vivax* multidrug resistance 1 (MDR1) copy number variation⁶¹.”

3. Line 205: And/or less frequent co-transmission? How does F_{ws} relate to MOI as separately determined?

We have revised the text to include the comment on co-transmission as follows:

Line 102 “...indicative of less frequent superinfection and/or co-transmission”

We have also added further details on the interpretation of the F_{ws} to the text as follows:

Lines 482-486 “Within-host infection complexity was assessed using the within-sample F statistic (F_{ws})^{52,53}, which estimates the fixation of alleles within each infection relative to the diversity observed in the total population on a scale from 0 to 1. Previous studies have demonstrated that an $F_{ws} \geq 0.95$ is highly indicative of clonal infection^{52,53}.”

4. Line 230: What does 1.81% SNPs difference mean here? Is that the median SNP difference % for comparisons that do not involve K2 isolates? Or including them?

We thank the reviewer for pointing this out. The original value presented indicated the median SNP difference % for comparisons that did include the K2 isolates. We have revised the text and now provide the median SNP difference % for comparisons with only a single representative of the K2 isolates; we believe that this is a more informative measure of the baseline population diversity.

Lines 129-132 “After excluding the imported isolates and using a single representative of the K2 strain, the median SNP-based nucleotide difference across Sabah was 11,333 (2.15% SNPs), demonstrating a moderately diverse underlying reservoir of infection despite the high prevalence of identical isolates.”

5. Line 238/Fig 4: Would the haplotype signatures on chromosomes 5 and 12 be clearer using an identity-by-descent-based metric (eg Isorelate/hmmIBD) rather than π ? π is a continuous statistic, with its mean value here determined by the proportion of comparisons between identical vs different haplotypes, but really what the authors are differentiating are genomic regions that show an

enhancement of virtually perfect identity among samples due to recent common ancestry, from those that do not. π is a less elegant way of getting at this, which could miss IBD blocks that are smaller or less frequent.

We thank the reviewer again for their comment. We have replaced the analyses using π with IBD data calculated using the hmIBD software. The chromosome 5 and 12 signals remained strong, and additional signals were identified. The hmIBD results are presented in Supplementary Fig 3 and Supplementary Table 3. The hmIBD methods are described in lines 489-495, and the results are described in lines 136-161.

6. Line 284: How are the estimates of F_{ST} and R_{sb} impacted by the high prevalence of the K2 clone in Sabah/Malaysia? Is a Weir and Cockerham estimator of F_{ST} appropriate here given unequal sample size (Malaysia \ll Thailand/Indonesia)? Are the MDR2 variants still significant in a comparison of Thailand to Indonesia, leaving out Malaysia?

As mentioned on lines 190-191, “The extensive homology among the K2 isolates obscured signal detection in Sabah, hence, analysis was undertaken with a single representative of this strain”.

Regarding the F_{ST} analysis with different sample sizes, based on evidence provided by Weir and Cockerham using simulations to illustrate the effects of sample size and other parameters (Evolution, 38(6), 1984, pp1358-1370), their estimator performs well in a wide range of conditions. According to the authors “Unlike the estimators in general use, the formulae do not make assumptions concerning numbers of populations, samples sizes, or heterozygote frequencies. As such, they are suited to small data sets and will aid the comparisons of results of different investigations.” We were highly stringent with our thresholds for defining high F_{ST} s, using a cut-off of ≥ 0.8 ; at these extremes these positions will be highly differentiated using any of the other F_{ST} measures.

The MDR2 variants were not significant in comparisons of Thailand versus Indonesia. We have revised lines 202-203 to clarify this: “Two nonsynonymous MDR2 variants (V43L and A603T) were among the top 0.5% of SNPs identified with R_{sb} analysis in comparisons against Sabah.” As mentioned in the discussion (lines 377-380), “..comparison with the Indonesian isolates, where high-grade CQR has been reported previously, may have been confounded by the high rate of local transmission⁶⁸, which may have rapidly reduced haplotype homozygosity surrounding any drug resistance alleles in this population”.

7. Assuming that cynomolgi here is assumed to be the ancestral rather than derived allele.

Yes, that is correct. We have clarified this in the text on line 231 “...ancestral (*P. cynomolgi*) relative to the derived allele.”

8. Figure 3: The rooted tree more clearly shows the K2 clone. Necessary to include the unrooted tree? That one appears redundant given that the other Malaysian clones besides K2 are visible in panel a.

Figure 3 has been revised to include the rooted tree only.

Reviewer #2

General comments

Thank you for the opportunity to review the manuscript entitled “Genomic analysis of Plasmodium vivax in Malaysia reveals selective pressure in a parasite population approaching elimination” by Auburn et al. This manuscript describes the population genetic analysis of 51 P. vivax parasite genomes from Sabah, Malaysia in comparison with genomic data from 104 P. vivax samples from each of western Thailand and Papua, Indonesia. The data reveal that just over half (54%) of the P. vivax samples from Malaysia are essentially genetically identical, perhaps representing a recent clonal expansion; and that genomic diversity reveals patterns of selection, possibly as a consequence of antimalarial drug use. Overall, this is a nice descriptive study of the P. vivax parasite population structure in a declining transmission context where chloroquine resistance (CQR) is relatively high. The major concern with the manuscript is the potential over-interpretation that signatures of selection are the consequence of drug pressure, given a lack of phenotypic or validation data. Nevertheless, the work represents a good description of P. vivax parasite population structure in a setting where transmission is declining that can inform potential challenges for elimination of P. vivax.

We thank the reviewer for their overall positive and constructive feedback. Please see our feedback in response to Reviewer 1’s General Comments, which apply here as well.

Major Comments

1. *I would encourage the authors to avoid potential conflation of “selection” and “clonal expansion” to infer that clonal expansion is a consequence of drug pressure.*
 - a) *The large proportion of “clonal” (K2) parasites is consistent with a recent clonal expansion (based upon ISA data).*

As per our response to Reviewer #1 comment 1, we have chosen to retain the term “clonal expansion” because the available evidence strongly supports this dynamic including the ISA data and, for the specific case of K2, the change in prevalence from 0% in 2010-12 to 29% in 2013, 55% in 2014 and 83% in 2015.

- b) *There is evidence of genetic loci (e.g., genomic regions on chromosomes 5 and 12), but these are found in both the K2 and K3 Malaysian populations. Thus, these regions are not likely to harbour variants that selectively contribute to the clonal expansion of the K2 lineage.*
- c) *Only dhps and the quadruple dhfr-ts haplotype differs between the K2 and K3 populations among known drug resistant loci tested; and, other regions under selection do not explain a possible “recent clonal expansion” of K2, especially since these mutations are enriched in the K3 parasite population.*
- d) *There is evidence of selection; but, I am not convinced that there is evidence that drug resistance is driving this lineage. For example, perhaps the K2 lineage is more easily able to be transmitted.*

In response to b) - d), we have now added further details to the main text to clarify that our investigation was not limited to the K2 strain since this was not the only highly divergent cluster

observed in Sabah. The extensive, “Cambodian-like” structure observed in Sabah extends beyond K2 into the K3 and mixed infections. We have also added a line clarifying that additional phenotypic data is needed to confirm that selective pressures have at least in part shaped the population structure.

Lines 318-323 “The Sabah *P. vivax* population exhibited a highly structured pattern, with divergence not just of the K2 strain, but also among the K3 and mixed infections. The extreme divergence in Sabah relative to the other Asia-Pacific populations mirrors that observed in *P. falciparum* in Cambodia as artemisinin-resistant founder populations emerged²⁹, suggesting that founder populations may already be emerging in response to selective pressures. However, additional phenotypic data is needed to confirm this hypothesis.”

Lines 340-348 “Investigation of polymorphisms previously associated with drug resistance in *P. vivax* revealed differences in the prevalence of several *DHPS* and *DHFR-TS* alleles between K2 and K3, potentially reflecting sulfadoxine-pyrimethamine pressure. The extensive homology in the K2 sub-population did not permit genome-wide investigations of differential selection and haplotype homozygosity to identify any novel selective determinants. Nonetheless, although the K2 sub-population was notable for its extreme genetic identity, the extensive divergence across Sabah was not restricted to this strain. Therefore, rather than limiting our investigation of drug-related selective pressure to the K2 strain, we undertook genome-wide tests of selection in the baseline Sabah population using a single K2 representative.”

e) *If drug resistance were the selective force driving expansion of the K2 parasite lineage, you would expect to see increases in the allele frequency of those markers over time, or association of specific variants with a drug resistant phenotype. You might also expect this lineage to be further increased in 2014, but peak seems to be 2013 by STR analysis.*

The peak prevalence of the K2 strain was not in 2013, but in 2015 (the last year investigated). As per our response to Reviewer #1 comment 1, the K2 strain underwent a rapid change in prevalence from 0% in 2010-12 to 29% in 2013, 55% in 2014 and 83% in 2015. This is now clarified in the main text (lines 267-269).

2. I would encourage the authors to provide more comprehensive information on polyclonal and monoclonal samples; how they were handled in the analysis; and to clarify the potential sources of bias in how polyclonal infections were handled.

a) *FWS is used to identify monoclonal infections. The authors should clarify the comparison used to derive the p value (0.0234, line 204) provided to support that monoclonal infections were enriched among Malaysian samples.*

The text has been revised (line 101) to clarify the use of Pearson’s chi-square test.

b) *The authors argue that more monoclonal infections equate to “less frequent superinfection”, but polyclonal infections can come from superinfection, co-transmission, relapse, or a combination thereof.*

We have revised the text as per Reviewer #1 comment 3.

c) ***Please specify how polyclonal infections were addressed in the analysis and how any potential bias toward relatedness was considered in the analysis.***

d) ***For example, in the STR analysis, only the dominant allele was used to create the MLG.***

In a previous *P. vivax* genomic study (Pearson *et al.*, Nat Gen, 2016 [27348299]), where we did not restrict analysis to monoclonal samples, we demonstrated proof of principle with the detection of several known drug resistance loci using haplotype-based signals of selection. In the current analysis we did not restrict our analyses to purely monoclonal samples ($F_{WS} \geq 0.95$) and instead used a less stringent threshold of $F_{WS} \geq 0.6$ to maintain a larger sample size. Nonetheless, we have repeated the haplotype-based analyses of selection with monoclonal samples ($F_{WS} \geq 0.95$) and provided the results in Supplementary Figure 4, demonstrating that there was no difference in the key trends. Details have been added to the text to clarify these points:

Lines 517-521 “Aside from the F_{WS} and F_{ST} tests, all genomic analyses were undertaken using the major allele call at heterozygous positions. IBD analysis was undertaken on monoclonal samples ($F_{WS} \geq 0.95$). The *Rsb* and *iHS* tests were undertaken on low complexity samples (defined as $F_{WS} \geq 0.6$), and additionally assessed with monoclonal samples ($F_{WS} \geq 0.95$). All other genomic analyses were conducted without F_{WS} filters.”

Lines 245-246: “Assessment of the *iHS* and of the *Rsb* tests using strictly monoclonal samples demonstrated no major differences in the key trends, confirming the accuracy of the results in the low complexity samples (Supplementary Fig. 4).”

The STR-based analyses of LD and allelic richness have been re-run using low complexity samples (defined as having a maximum of 1 multi-allelic locus) to assess the potential impact of errors in MLG reconstruction. The data has been added to Supplementary Tables 10 and 11, and confirms that the key trends remain the same as in the unfiltered data. Details on the additional analyses and their results have been added to the text:

Lines 540-544 “Apart from measures of polyclonality and MOI, only the predominant allele at each locus in each isolate was included in analysis⁶⁸. *Rs* and LD analyses were conducted on all infections and additionally in low complexity samples (maximum 1 multi-allelic locus) to assess the potential impact of MLG reconstruction errors.”

Lines 284-285 “The trends in *Rs* and LD remained the same when analysis was restricted to low complexity infections (Supplementary Tables 10 and 11).”

3. *Analysis to quantify parasite genetic relatedness (done by neighbour-joining and PCoA) could be enhanced by testing for identity by descent (IBD) analysis that can quantify both the proportion of the genome that is related as well as the length of the chromosomal segments that are identical by descent to infer relatedness.*

As per our response to Reviewer #1 comment 5, we have included IBD analysis using the hmmIBD software.

4. *Interpretation that K2 infections were increased among student population, but perhaps this is an age-related effect (i.e., immunity) since students are likely enriched among younger aged*

individuals.

We have revised the text to comment on the possibility of immune-related factors concerning the increased risk of K2 carriage amongst students:

Lines 393-394 “This finding might reflect factors related to the immune status of this particular age group who had a median age of 15 years.”

Minor Comments

1. Indicate that 3 parasites are “imported”: (a) state how imported was determined/defined (i.e., travel history); (b) identify these imported infections on analysis (i.e., are they distinct in the PCoA and ADMIXTURE analysis); and, (c) do they align with other *P. vivax* populations consistent with importation? For example, there seem to be at least two Malaysian samples that cluster with the Thailand or Indonesian parasites (Figure 2a), are these the imported parasites or parasites from the “west coast” rather than “Kudat”?

(a) The text has been revised to clarify the definition of the putatively imported cases:

Lines 117-118 “and 3 (6%) cases defined as putatively imported on the basis of ancestry ranging from 80 to 100% to K1 or K4 (Figure 2c).”

(b) Figures 2 and 3 have been revised to include annotation of the putatively imported cases.

(c) Additional details have been added to the text in the Figure 3 legend describing the alignment of the putatively imported cases to each population represented as follows:

Lines 770-776 “Three putatively imported cases presenting in Sabah are annotated with black stars. One of the putatively imported infections (PY0045-C) aligned with the Papua Indonesian isolates, suggestive of importation from this region, whilst the other two cases aligned between Papua Indonesia and Sabah (PY0004-C), and close to Thailand (PY0120-C), presumably reflecting importation from regions not represented by the available sample set.

2. For PCoA, please specify how much of the variance of the data is explained by the components.

The variance explained by each of the components is now indicated in the legend of Figure 2 as follows:

Lines 756-757 “Principal Components 1-4 reflect 17.6%, 11.7%, 3% and 1.3% of the variance respectively.”

3. Define what is meant by “high grade” and “low grade” CQR.

We have revised the text on lines 75-79 to define the degree of CQR apparent in Thailand and Indonesia, providing the relevant references.

4. You indicate the divergence between K2 and K3 is greater than the divergence between K1 and K4 (by F_{ST}), is that what you would expect given population fragmentation within Malaysia?

Yes, this pattern has been observed in a separate study highlighting a highly structured *P. falciparum* population in Cambodia. In this previous analysis the pairwise F_{ST} between two Cambodian subpopulations ($F_{ST} = 0.38$) was higher than that for two distinct countries, Thailand and Ghana ($F_{ST} = 0.16$) (Miotto et al., Nat Gen, 2013 [PMC3807790]).

5. Infections from PCD, comment on how these may (or may not) represent the parasite population given so few cases. Could reservoir infections be different and clonal parasites be found only among those who are symptomatic?

Using microsatellite-based data on passively and actively detected *P. vivax* samples from the moderately high endemic setting of Timika, Papua Indonesia, we previously observed similar genetic diversity between the two reservoirs, but polyclonal infections were more prevalent among the actively detected cases (Pava et al., 2017, AJTMH [29016343]). However, there have not been any studies comparing the genetic make-up of *P. vivax* isolates in a low endemic setting such as Sabah. We have added several lines to the text commenting on these points:

Lines 406-413 “Aside from imported infections, it will be important to determine the contribution of a hidden reservoir of asymptomatic or subpatent infections in maintaining local diversity, and whether the genetic structure of these infections is comparable to the clinical cases. A genetic study of actively and passively detected *P. vivax* cases from Timika, Papua Indonesia, revealed similar genetic diversity between the two reservoirs, although a higher frequency of polyclonal infections in the actively detected cases³⁹. However, there are no data available on the genetic make-up of passively versus actively detected *P. vivax* in low-endemic settings.”

6. Include reference for R_{sb} measure of cross-population extended haplotype homozygosity (Tang, PMID: 17579516).

A reference has been added to the text.

7. The descriptions regarding analysis for both known drug resistance loci and those identified through selection analysis are quite wordy and difficult to follow. Perhaps including a summary table with the high level findings would help the reader.

We have revised the text in several of these sections to clarify. A high-level summary of resistance loci with previously demonstrated associations in *P. vivax* is presented in a summary table (Table 1), and we have compiled all the high-level findings on signals of selection to detect new candidates (F_{ST} , R_{sb} and iHS) into a single, more concise Supplementary Table (Supplementary Table 4).

8. Supplemental Figure 1a: there are no regions on the map that are light grey (representing “*P. vivax* free”); and, no regions indicated by striped grey (representing “Unstable transmission and high Duffy negativity”).

This is a generic legend that is used in all Malaria Atlas Project maps but we appreciate that the “*P. vivax* free” and “Unstable transmission and high Duffy negativity” references are not applicable to this particular map and have thus removed them from Supplementary Figure 1a.

9. *Supplemental Figure 1b: state that “incidence of *P. vivax* has declined substantially over the past decade”, but would be better to qualify that based upon Supplemental Figure 1b showing for cases reported that there was not much difference between 2003 and 2015 endpoints, and that there was evidence of an increase in cases between 2005 and 2010, followed by a decrease in cases from 2010 to 2015. It would be valuable to include prevalence or incidence data for these periods, if available, for context. Do these trends correspond to the decline in polyclonal infections over time?*

Using details on the number of cases reported and information available from the Malaysian Department of Statistics, we have approximated the annual *P. vivax* incidence per 1000 persons in Sabah in 2000, 2010 and 2015 (2005 estimate not available). These details have been added to the legend of Supplementary Figure 1 as follows:

Lines 960-962 “Based on population size data provided by the Malaysian Department of Statistics, the annual *P. vivax* incidence (per 1000 population) was approximately 0.78 in 2000, 0.32 in 2010 and 0.02 in 2015.”

There was a strong, positive correlation between the annual number of reported *P. vivax* cases and the annual percentage of polyclonal infections during the years tested (2010-2014). Additional details have been added to the text as follows:

Line 276-278 “...and this was associated with a strong, positive correlation between the annual number of *P. vivax* cases and the proportion of polyclonal infections (Pearson’s correlation, $r=0.98$, $p=0.004$).”

10. *More information about the details of the figures should be provided in the legends. For example, Figure 1 does not describe the information in the figure adequately and only provides a high level summary.*

Additional details have been provided in the legends where the word limit permits.

11. *Clarify comment: “...continual flux was observed in the genetic make-up of the parasite population, including several rapidly emerging clonal expansions”, given there was one clonal population that was consistent with rapid emergence.*

The text has been revised for clarity as follows:

Lines 306-307 “A continual flux was observed in the genetic make-up of the parasite population, including several small and one large, rapidly emerging clonal expansion.”

12. *The authors speculate that heavy drug pressure (and other selective forces) have “bottlenecked” the population, leaving the fittest isolates remaining. There is no evidence that the “fittest” have survived...and if they are indeed due to drug resistance than they may be survivors because of that phenotype. Many drug resistant parasites are actually “less fit” than their wild-type counterpart.*

We have revised the text as requested omitting the wording “leaving the fittest isolates remaining” from line 310.

13. *The authors postulate that drug pressure has shaped the population...would have been nice to see that drug resistance or other loci are enriched in the K2 population that has expanded. It is fine to speculate, but just be sure to qualify statements appropriately.*

Please see the response to Reviewer #2 major comment 1b, clarifying that the investigation of local selective pressures was not limited to the K2 strain as extensive divergence was also observed amongst the K3 and mixed ancestry infections, and clarifying that we did in fact see differences at the DHPS and DHFR-TS loci between K2 and K3.

14. *“Intense drug selection is an inevitable consequence of the final states of elimination, providing an environment in which only the most drug-resistant infections survive”, while this may be the case in some geographic settings (depending upon drug pressure etc), it is not necessarily “inevitable”.*

Evidence from areas where infectious diseases have been reduced by applying intense drug pressure almost all highlight an increasing prevalence of drug resistant parasites and this is particularly apparent for malaria. However we have toned down the text to use the word “highly likely” instead of “inevitable” (line 314).

15. *MDR2 is focused on as a possible driver...in Ch12 region, but how does this relate to population structure of K2? I would be careful not to over interpret the identification of new candidates without any phenotypic or validation information. Fine to put them out there, and discuss how they might play a role, but not sure there is strong evidence to suggest they are playing a role in the population structure observed.*

Please see the response to Reviewer #2 major comment 1b, clarifying that the investigation of local selective pressures was intentionally not limited to the K2 strain.

16. *Caution around the interpretation of students, as age may be the important factor and possibly related to changes in immunity. Arguments about risk populations might be addressed by spatial or temporal data, should this be available.*

The text has been revised as per Reviewer #2 Major comment 4.

REVIEWERS' COMMENTS:

Reviewer #1 (Remarks to the Author):

I am satisfied with the authors' responses to my comments and those of the other reviewer. I support the publication of this manuscript.

Reviewer #2 (Remarks to the Author):

Overall, I believe that the authors have addressed all the major issues.

I would still encourage care with the term 'clonal expansion' that remains in the manuscript. As pointed out by both reviewers, there are alternate explanations for detecting the dominance of a clonal population given the massive reductions in transmission thus population bottlenecks. Yet, the abstract still includes the strong statement: "54% of the Sabah isolates had identical 39 genomes, reflecting a rapid clonal expansion." Simply changing the term "reflecting a rapid clonal expansion" to "consistent with a clonal expansion" would be helpful. There is a more balanced statement (line 258) that includes the composite of bottleneck and selection, but then in the discussion again is re-assertion of clonal expansion (line 284) or bottlenecked under selection (line 286-287). There remains conflation around population restriction (bottleneck) and emergence of drug resistant lineage (selection) that should be clarified by the authors.

Reviewer #1

I am satisfied with the authors' responses to my comments and those of the other reviewer. I support the publication of this manuscript.

We thank the reviewer again for their time in reviewing the manuscript.

Reviewer #2

General comments

Overall, I believe that the authors have addressed all the major issues.

I would still encourage care with the term 'clonal expansion' that remains in the manuscript. As pointed out by both reviewers, there are alternate explanations for detecting the dominance of a clonal population given the massive reductions in transmission thus population bottlenecking. Yet, the abstract still includes the strong statement: "54% of the Sabah isolates had identical 39 genomes, reflecting a rapid clonal expansion." Simply changing the term "reflecting a rapid clonal expansion" to "consistent with a clonal expansion" would be helpful. There is a more balanced statement (line 258) that includes the composite of bottleneck and selection, but then in the discussion again is re-assertion of clonal expansion (line 284) or bottlenecked under selection (line 286-287). There remains conflation around population restriction (bottleneck) and emergence of drug resistant lineage (selection) that should be clarified by the authors.

We thank the reviewer again for their time and constructive feedback. We have made several revisions to the text to soften the statements concerning clonal expansion and selection as follows:

Lines 39-40 now read "54% of the Sabah isolates have identical genomes, consistent with a rapid clonal expansion."

Lines 293-296 now read "Whilst some of these changes may have resulted from demographic or neutral processes, others may reflect more concerning adaptive changes as the parasite population is bottlenecked under heavy pressure from antimalarial drugs and other selective forces."

Lines 293-296 now read "However, additional phenotypic data are needed to confirm this hypothesis as the extensive divergence in Sabah may also reflect extensive bottlenecking alone."